# Mapping expectancy-based appetitive placebo effects onto the brain in women

Iraj Khalid[1], Belina Rodrigues [1], Hippolyte Dreyfus[1], Solène Frileux[1], Karin Meissner [2,3], Philippe Fossati[1,4], Todd Anthony Hare [5] & Liane Schmidt [1] ✉

Suggestions about hunger can generate placebo effects on hunger experiences. But, the underlying neurocognitive mechanisms are unknown. Here, we show in 255 women that hunger expectancies, induced by suggestion-based placebo interventions, determine hunger sensations and economic food choices. Functional magnetic resonance imaging in a subgroup ($n = 57/255$) provides evidence that the strength of expecting the placebo to decrease hunger moderates medial prefrontal cortex activation at the time of food choice and attenuates ventromedial prefrontal cortex (vmPFC) responses to food value. Dorsolateral prefrontal cortex activation linked to interference resolution formally mediates the suggestion-based placebo effects on hunger. A drift-diffusion model characterizes this effect by showing that the hunger suggestions bias participants' food choices and how much they weigh tastiness against the healthiness of food, which further moderates vmPFC−dlPFC psychophysiological interactions when participants expect decreased hunger. Thus, suggestion-induced beliefs about hunger shape hunger addressing economic choices through cognitive regulation of value computation within the prefrontal cortex.

A fundamental aspect of human cognition is the ability to extract patterns from noisy sensory information to form prospective beliefs (expectations). Statistical frameworks propose that the brain achieves such integration through a computational process that continuously updates prospective beliefs based on prior beliefs and new belief-confirming or disconfirming evidence[1]. This idea has been shared and challenged for centuries[2,3]. Interestingly, the combination of prospective beliefs with sensory information coming from the outside in the form of verbal suggestions has been shown to mediate placebo effects, which are a famous example of mind-brain-body interactions[4–7].

Placebo effects are much known from clinical research as a source of noise, which can obscure the benefits of an active treatment, which is then commonly inferred to be non-effective. However, basic

research in cognitive neuroscience has provided evidence for active neural and cognitive processes under placebo effects on bodily responses and a person's judgment of experience[4–10]. These interventions combine the administration of an inactive substance (a placebo) with a verbal suggestion about its effectiveness, which are sometimes, but not always, reinforced by conditioning to generate acute placebo effects in the laboratory. Much of this previous basic science work has used aversive outcomes, such as pain combined with functional magnetic resonance imaging (fMRI), to map placebo hypoalgesia onto the brain (for review[11]).

But, placebos combined with verbal suggestion can also affect appetitive outcome measures. Research on consumption behavior has shown for example that the taste of more expensive wines is experienced as more pleasant and is encoded more strongly in the brain's

[1]Sorbonne University, Institut du Cerveau—Paris Brain Institute – ICM, INSERM, CNRS, APHP, Hôpital de la Pitié-Salpêtrière, Paris, France. [2]Institute of Medical Psychology, Medical Faculty, LMU Munich, Munich, Germany. [3]Division of Health Promotion, Faculty of Applied Natural Sciences and Health, Coburg University of Applied Sciences and Arts, Coburg, Germany. [4]Adult Psychiatry Department, APHP, Hôpital de la Pitié-Salpêtrière, Paris, France. [5]Zürich Center for Neuroeconomics, Department of Economics, University of Zürich, Zürich, Switzerland. ✉e-mail: liane.schmidt@icm-institute.org

valuation system than identical wines suggested to be less expensive[12,13]. Similar findings showed that verbal suggestions about caloric ingredients or the homeostatic effectiveness of an inactive substance can influence the release of satiety signaling hormones[14], interoceptive hunger experiences[15], and digestion-related autonomous nervous system responses[16]. Similar to placebo effects in aversive domains, such appetitive placebo effects are mediated by a participant's positive expectations about consumption or effectiveness[17].

Despite the ample evidence for appetitive placebo effects on the behavioral side, there is, however, no direct empirical evidence for when, where, or how, in the brain, a placebo intervention that combines the administration of an inactive substance with a verbal suggestion about its effectiveness influences the experience of appetitive interoceptive outcomes, such as hunger and associated economic behavior. Addressing these questions is important, because it provides an opportunity to understand the effects of higher-order cognitive factors, such as prognostic beliefs, more broadly and how the brain integrates them to make inferences about signals from the body and shape economic behavior that addresses interoceptive and exteroceptive signals.

To address this question, this study builds on literature from decision neuroscience that has shown that a person's goals and internal bodily states can affect economic choice behavior. More specifically, generic models of economic choice propose that decision-making involves valuation and action selection (decision) stages[18]. During valuation, various attributes of alternative choices are weighed and integrated into stimulus values that approximate hidden preferences and are then compared during the decision stage to select the most preferred alternative (i.e., with the higher stimulus value). The underlying neural mechanisms have been shown to involve regions of the brain's valuation system, such as the ventromedial prefrontal cortex (vmPFC), and cognitive regulation system, such as the dorsolateral prefrontal cortex (dlPFC)[19-29]. Furthermore, a person's goals can influence the valuation and decision stages and the underpinning brain responses through cognitive regulation in the form of attentional filtering and value modulation[19,20,30,31] of relevant information. However, it is unknown whether verbal suggestions about the effectiveness of a placebo on interoceptive states can generate such cognitive regulation at the valuation and/or at the decision stages of economic choices. Moreover it is unknown, where and how in the brain such potential effects take place.

Given that placebos together with suggestion can affect hunger experiences and that economic decision-making addresses hunger, we used a placebo intervention that involved the administration of an identical drink (water), together with the verbal suggestion that the water either increased or decreased hunger. We also tested a control group that drank the water without suggestions about its potential effect on hunger. We hypothesized that the placebo intervention would induce prognostic beliefs about the effectiveness of the drink's effect on hunger and through them, affect the experience of hunger, dietary decision-making, and its cognitive regulation. We combined fMRI in a subgroup of the two hunger suggestion groups (e.g., decreased vs. increased) with a dietary decision-making task performed by participants in all groups, and fitted economic choices and reaction times obtained from all participants to a drift diffusion model (DDM)[32,33]. This approach allowed us to assess where in the brain, when during economic choice formation, and how the placebo intervention affected hunger-addressing economic behavior.

In accordance with our hypotheses, a main effects test conducted across the entire sample found that the placebo intervention generated hunger expectations that were stronger in the hunger suggestion groups than in the control group, and then determined how hungry participants felt at the end of the experiment. The strength of expecting the water to decrease hunger moderated activation of the medial prefrontal cortex at the time of food choice. Consistent with

these effects, participants in the increased-hunger suggestion group valued food more highly and displayed stronger vmPFC activation in response to food value than participants in the decreased-hunger suggestion group. The suggestion-based placebo effect on hunger ratings was further formally mediated by the recruitment at time of food choice of a dlPFC region that was associated with interference resolution. Drift diffusion modeling then revealed that the hunger suggestion influenced the dynamics of the decision stage. Participants in the increased-hunger suggestion group were initially more biased toward yes food choices, and considered the tastiness of the food more strongly. In contrast, participants in the decreased-hunger suggestion group were less initially biased toward yes, and considered the healthiness of the food more strongly during the decision stage. The relative drift weights of these two food attributes on the accumulation of evidence toward a yes/no food choice then moderated how strongly the vmPFC and the interference-resolution associated dlPFC regions of interest interacted during choice formation.

## Results

### Expectancy ratings

We first checked whether the placebo intervention was successful in generating prognostic beliefs about the drink's effectiveness on hunger and whether these expectations were different in their strength relative to the no-suggestion control group. Note, all groups used the same scale from 1 (no expected effect) to 10 (maximum expected effect), but the framing of the scale was different. The decreased expectancy group was asked to rate how much they expected the drink to decrease their hunger. The participants in the increased expectancy group were asked to rate how much they expected the drink to increase their hunger. The control group was asked to rate how much they expected the drink to affect their hunger. Thus, we inferred that the intervention aimed at inducing different beliefs should not vary in strength between the two hunger suggestion groups, but rather between the hunger suggestion and no hunger suggestion control groups.

A mixed effects linear regression analysis of expectancy ratings showed a significant and positive intercept ($\beta = 6.1$, SE = 0.8, $p = 6.7e-13$, 95% CI [4.5−7.7]), indicating that these ratings across the entire sample were different from zero in their magnitude ($n = 255$). We then found a significant negative main effect of group ($\beta = -0.75$, SE = 0.16, $p = 6.6e06$, 95% CI [−1.1 to −0.43]), controlled for BMI, which had a non-significant effect ($\beta = 0.04$, SE = 0.03, $p = 0.18$, 95% CI [−0.02 to 0.1]). Posthoc independent t-tests then showed that the two hunger suggestion groups did not differ in how much they expected the drink to effect their hunger (mean expectancy rating$_{decreased}$ = 5.6 ± 0.2 [$n = 88$] vs mean expectancy rating$_{increase}$ = 5.3 ± 0.2 [$n = 84$], t(169) = 1.12, $p = 0.26$, two-tailed, BF$_{10}$ = 0.29). However, the strength of the hunger expectations significantly differed in both hunger suggestion groups from that of the control group (control < decreased: t(169) = −4.6, $p < 0.001$, Cohen's $d = -0.7$, BF$_{10}$ = 1863; control < increased t(165) = −3.6, $p < 0.001$, Cohen's $d = -0.6$, BF$_{10}$ = 56, two-tailed, Fig. 1a).

These findings confirm that the placebo intervention successfully induced stronger expectations about the drink's effectiveness on hunger in the two suggestion groups than in the no-suggestion control group. Moreover, the more participants in the decreased-hunger suggestion group ($n = 88$) expected the drink to efficiently decrease their hunger, the less their hunger increased over the course of the experiment (Pearson's $R = -0.24$, $p = 0.03$, Fig. 1c). By contrast, this correlation was non-significant among participants in the increased-hunger suggestion group ($n = 84$, Pearson's $R = -0.08$, $p = 0.44$).

### Expectancy encoding in the brain at the time of food choices

We then investigated where in the brain these prognostic beliefs were encoded within each suggestion group of the fMRI sample. We found

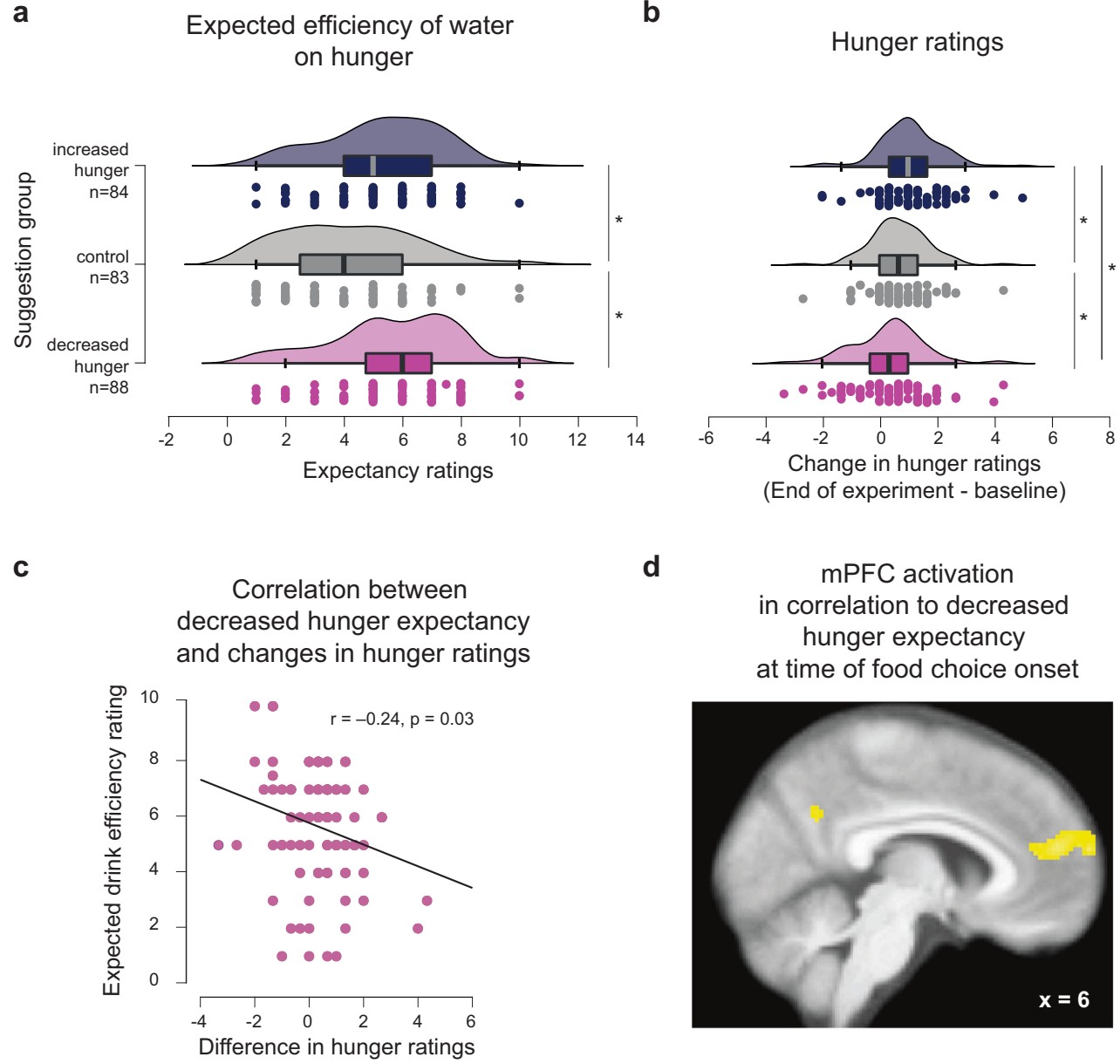

**Fig. 1 | Placebo effects on expectancy ratings.** All boxplot graphs display the 95% confidence intervals with boxes indicating the interquartile range from Q1 25th percentile to Q3 75th percentile. The gray and black lines indicate medians and whiskers range from minimum to maximum values and span 1.5 times the inter- quartile range. The dots in the rainclouds above each boxplot correspond to individual participants. Light red corresponds to the decreased hunger suggestion group, gray to control group and blue to the increased hunger suggestion group. **a** expectancy ratings following the question: How much do you expect the drink to decrease/increase/affect your hunger?, and **b** for the change in hunger ratings from baseline to the end of the experiment. The hunger ratings were averaged for each participant between three composite ratings addressing hedonic (How pleasant would it be to eat right now?), homeostatic (How much could you eat right now?), and subjective general hunger ratings (How hungry are you right now?). Dots correspond to individual ratings in each group. *$p < 0.05$. $P$-values were obtained with two-sampled, two-tailed t-tests. **c** Pearson's correlation between expectancy ratings and the change in hunger ratings from baseline to the end of the experi- ment. Each dot corresponds to a participant in the decreased-hunger suggestion group. The $p$-value indicates a two-tailed difference from zero correlation. **d** Statistical parametric maps of significant group level, random effect moderation of brain activation at the time of food choice by expectations about the drink's effectiveness to decrease hunger ($n = 28$). Voxels in yellow are displayed for visualization purposes at an uncorrected threshold of $p < 0.001$, with an extended threshold of $k = 44$ voxels, which corresponded to $p_{FWE} < 0.05$ cluster-wise cor- rection for family-wise errors across the whole brain. Activation was taken at the local maxima (MNI $x = 6$) on the sagittal slice, showing the extent of the activation in mPFC from the superior frontal gyrus to the anterior cingulate cortex. SPMs are superimposed on the average anatomical brain image. Source data and exact $p$ values are provided as a Source Data file.

that, at time of choice, the strength of expecting the drink to decrease hunger correlated significantly with activation of the superior frontal part of the medial prefrontal cortex (mPFC) extending into the anterior cingulate cortex (MNI = [2, 58, 22], $p_{FWE} < 0.05$, family-wise error cor- rected on the cluster level, Fig. 1d, SI Table 1). This finding was sig- nificant only for the participants assigned to the decreased-hunger suggestion group. The more these participants expected the water to decrease their hunger ($n = 28$), the more the mPFC was activated at the time of the food choice. No significant moderation of choice-related brain responses was observed for the increased-hunger suggestion group ($n = 29$) after correction for multiple comparisons at the cluster level across the entire brain.

## Suggestion-based placebo effects on hunger experiences

Consistent with the effects on prognostic beliefs about hunger, we conducted main effects tests across the entire sample ($n = 255$) by fitting a linear mixed effects model (LME) to hunger ratings. Note, these hunger ratings corresponded to an average between subjective ratings of three aspects of hunger for each participant: hedonic (how pleasant would it be to eat right now?), homeostatic (how much could you eat right now?), and subjective (how hungry are you right now?).

The LME showed a significant main effect of measurement time ($\beta = 0.71$, SE = 0.07, $p = 1.7e-21$, 95% CI [0.57–0.85]), a significant main effect of group ($\beta = -0.34$, SE = 0.15, $p = 0.02$, 95% CI [−0.64 to −0.04]), and a significant group by measurement time interaction ($\beta = 0.37 \pm 0.08$, $p = 2.1e-05$, 95% CI [0.20–0.54]). These effects were controlled for confounding due to BMI, which had a significant negative effect on hunger ratings ($\beta = -0.03$, SE = 0.01, $p = 0.03$, 95% CI [−0.07 to −0.004]). They were also non-different between behavioral pilot sample and the smaller subset of participants, who underwent the experiment during fMRI (SI section 1, SI Fig. 1).

Post-hoc t-tests then indicated that participants were hungrier at the end of the experiment than at the beginning in all groups (baseline vs end of the experiment: $t(87)_{decreased} = -2.53$, $p = 0.01$, Cohen's $d = -0.3$; $(t(83)_{increased} = -8.99$, $p < 0.001$, Cohen's d = −0.98; and $t(82)_{controls} = -6.21$, $p < 0.001$, Cohen's $d = -0.68$ paired two-tailed t-test; Fig. 1b).

In accordance with the main effect of group, participants in the increased-hunger suggestion group reported being hungrier at the end of the experiment than those in the control group ($t(165) = 2.37$, $p = 0.02$, Cohen's $d = 0.37$, $BF_{10} = 2.2$, two-tailed) and decreased-hunger suggestion group relative to baseline ($t(170) = -4.08$, $p < 0.001$, Cohen's $d = -0.62$, $BF_{10} = 289.3$, two-tailed) (Fig. 1b). On the contrary, participants in the decreased-hunger suggestion group reported being less hungry at the end of the experiment relative to baseline than the control group ($t(169) = -1.99$, $p = 0.04$, Cohen's $d = -0.31$, $BF_{10} = 1.03$, two-tailed).

## Suggestion-based placebo effects on food valuation

We then tested the effects of the placebo intervention on hunger-addressing value-based decision-making and related brain responses. A linear mixed effects model fitted to stimulus value ratings (i.e., how much participants wanted to eat the food items) found a main effect of group ($\beta = 0.10$, SE = 0.03, $p = 0.003$, 95% CI [0.03–0.16]), controlled for BMI, which had a borderline significant effect on stimulus value ratings ($\beta = 0.01$, SE = 0.006 $p = 0.05$, 95% CI [−0.0002 to 0.02]).

Post-hoc t-tests showed that participants in the decreased-hunger suggestion group ($n = 88$) assigned less value to food stimuli in terms of how much they wanted to eat the different snack food items than participants in the control group ($n = 83$, $t(169) = -3.03$, $p = 0.003$, Cohen's $d = -0.44$, $BF_{10} = 10.9$, two-tailed) and increased-hunger suggestion group ($n = 84$, $t(170) = -2.92$, $p = 0.004$, Cohen's $d = -0.45$, $BF_{10} = 8.1$, two-tailed), with no difference between the increased suggestion and control groups ($t(165) = -0.24$, $p = 0.81$, Cohen's $d = -0.04$, $BF_{10} = 0.2$; Fig. 2a).

A multilevel general linear regression model fitted to stimulus value ratings then showed a positive prediction by the tastiness of food across all three groups ($\beta_{decreased} = 0.54 \pm 0.03$, $t(87)) = 16.42$, $p < 2.22e-16$; $\beta_{increased} = 0.67 \pm 0.03$, $t(83) = 24.64$; $p < 2.22e-16$; $\beta_{controls} = 0.67 \pm 0.03$, $(t(82) = 20.3$, $p < 22.22e-16$, one-sample t-tests, SI Tables 2–4). Importantly, the effect of tastiness on food valuation was much stronger in the increased-hunger suggestion group than in the decreased-hunger suggestion group ($t(170) = -2.81$, $p = 0.005$, Cohen's $d = -0.46$, two-sample, two-tailed t-test, SI Table 5) and between the decreased-hunger suggestion and control groups ($t(169) = -2.92$, $p = 0.004$, Cohen's $d = -0.47$, two-sample, two-tailed t-test, SI Tables 6–8).

We further tested if stimulus value ratings correlated to calorie density. To this aim dietary decision making trials were split into low

and high calorie level trials using a median split of the calorie density values that were provided by the food stimuli database (see also SI table 11). Stimulus value ratings correlated significantly more with calorie density in high- (average Pearson's $R = 0.17$) than in low- (average Pearson's $R = -0.12$) calorie level trials ($z = 3.25$, $p = 0.0012$, two-tailed Fisher's r-to-z transformation, $t(254) = -16.22$, $p < 0.001$, paired-sample two-tailed t-test). This finding indicated that participants preferred high caloric food more than low caloric food (SI section 2.1). To test if this preference for high calorie food differed between groups, a linear mixed effects model revealed a small, borderline non-significant main effect of calorie level (high vs. low) on stimulus value ratings ($\beta_{calorie\ level} = 0.02 \pm 0.01$, $t(506) = 1.95$, $p = 0.052$, 95% CI [2.20e-4 to 0.05]) and a significant main effect of group ($\beta_{group} = 0.09 \pm 0.03$, $t(506) = 2.71$, $p < 0.001$, 95% CI [0.02–0.15]). There was no significant interaction calorie level (high vs. low) by group on food valuation. Post-hoc t-tests showed that participants in the increased-hunger suggestion group preferred both the low- ($SV_{low} = 2.23 \pm 0.04$) and the high-calorie foods ($SV_{high} = 2.30 \pm 0.06$) more than participants in the decreased-hunger suggestion group ($SV_{low} = 2.08 \pm 0.05$, $t(170) = 2.31$, $p = 0.02$, two-sample two-tailed t-test; $SV_{high} = 2.10 \pm 0.05$, $t(170) = 2.63$, $p = 0.009$, two-sample two-tailed t-test, Fig. 2b). Similarly, participants in the control group chose both low- ($SV_{low} = 2.27 \pm 0.05$, $t(169) = 2.69$, $p = 0.008$, two-sample two-tailed t-test) and high-calorie foods ($SV_{high} = 2.33 \pm 0.06$, $t(169) = 2.89$, $p = 0.004$, two-sample two-tailed t-test) more than those in the decreased-hunger group (Fig. 2b). Note, these effects were not interacting with the tastiness of food (see SI section 2.2).

## Suggestion-based placebo effects on valuation-related brain responses

The observed behavioral effects of suggestion on valuation were underpinned by stronger activation of the ventromedial prefrontal cortex (vmPFC), nucleus accumbens, posterior cingulate cortex, bilateral posterior insula, and precuneus in response to stimulus value ratings. Activation of these brain regions correlated more strongly with the food stimulus value in the increased-hunger suggestion group than the decreased-hunger suggestion group ($p_{FWE} < 0.05$, family-wise error corrected at the cluster and voxel level, Fig. 2c, d, SI Table 9). This difference was specific for the encoding of stimulus values. No differences between the two hunger suggestion groups were found for the encoding of tastiness or healthiness in the brain (see SI section 3, SI Tables 10, 11).

## Brain mediators of suggestion-based placebo effects on hunger

We next tested for formal neural mediators of observed suggestion-based placebo effects on hunger. We reasoned that if the hunger suggestion affects hunger via suggestion-consistent attentional mechanisms, brain regions associated with cognitive regulation such as attentional filtering should mediate such an effect. Attentional filtering can be measured by interference resolution during a Stroop effect. We thus first used a localizer task such as the Multi-Source-Interference task (MSIT) to identify the brain regions recruited when participants allocated attentional resources to solve interference from task-irrelevant information and to filter task-relevant information (SI section 6). Three regions, the dACC, dlPFC, and insula, were more strongly activated during incongruent than congruent blocks of trials (SI table 12). We then used this activation to functionally define regions of interest (ROIs), and tested whether the activation of the conflict resolution-related ROIs at the time of food choice during the dietary decision-making task mediated the behavioral placebo effects on hunger ratings. The results showed mediation to be significant for voxels located in the dlPFC ROI (Bonferroni corrected path a * b, $p = 0.008$), with a non-significant total effect of hunger suggestion on the hunger ratings after controlling for the mediator (path c': beta = 0.21, SE = 0.17, $p = 0.26$,

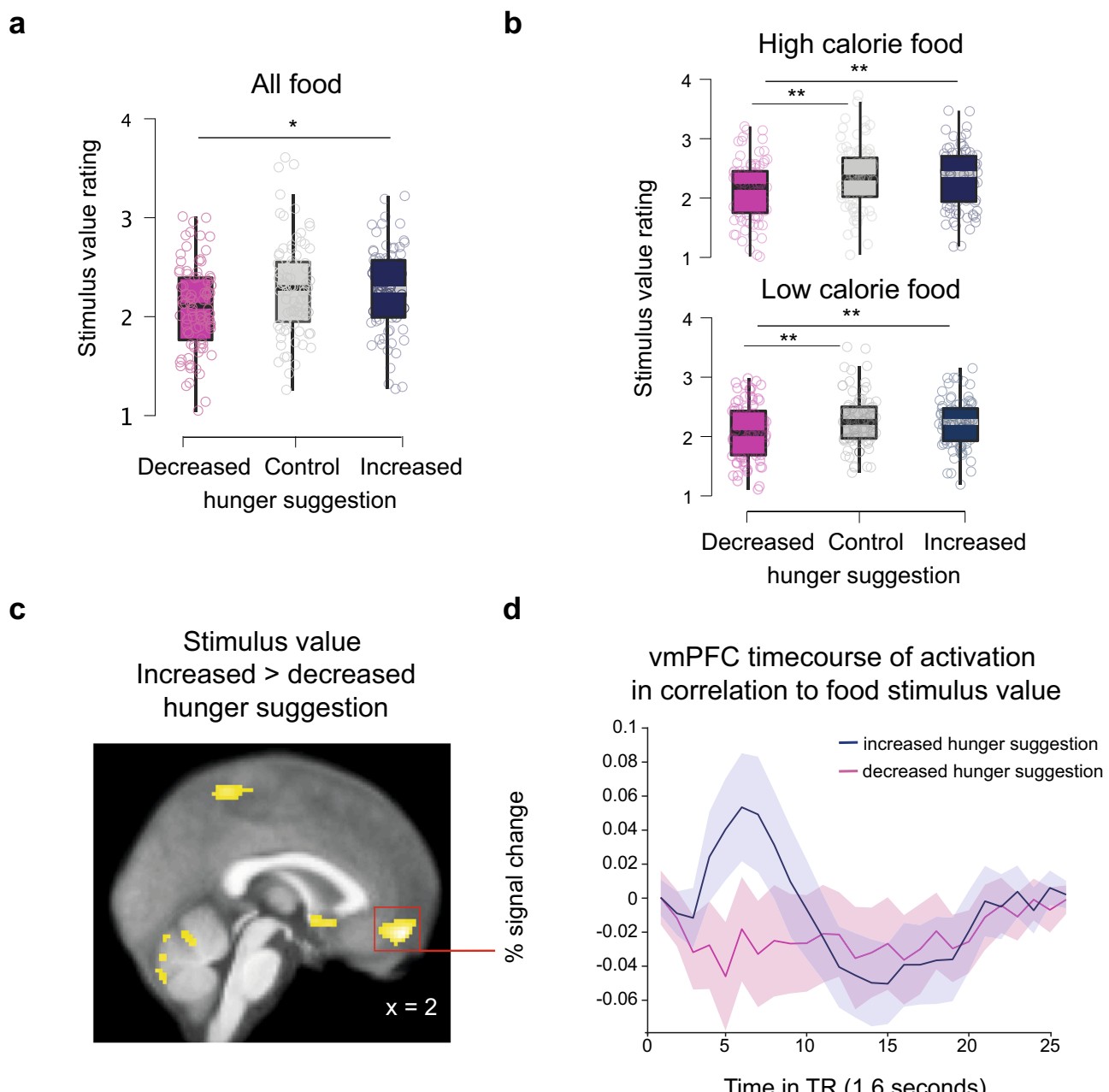

**Fig. 2 | Suggestion-based placebo effect on the valuation stage of dietary decision-making.** Behavioral placebo effect on: **a** food valuation across all trials and **b** food valuation split down into low and high calorie food choice trials. All boxplots display the 95% confidence intervals of the stimulus values (i.e., How much do you want to eat this food? given on a 4-point Likert scale) assigned to food during the dietary decision-making task for all three groups ($n = 255$ participants). Boxes correspond to the interquartile range from Q1 25th percentile to Q3 75th percentile. The gray and black lines indicate medians and whiskers range from minimum to maximum values and span 1.5 times the interquartile range. The dots correspond to individual participants. Light red corresponds to the decreased hunger suggestion group, gray to the control group, and blue to the increased hunger suggestion group. *$p < 0.05$, **$p < 0.01$. *P*-values were obtained with two-sampled, two-tailed t-tests between groups, and exact p-values are provided in the

source data file. **c** Neural placebo effect on food valuation in $n = 57$ participants. Statistical parametric maps (SPMs) display the contrast in brain activation between the increased versus decreased hunger suggestion group in response to the stimulus value at the time of the food choice. The significant voxels in yellow are corrected for multiple comparisons using family-wise error correction on voxel and cluster level ($p_{FWE} < 0.05$) and are superimposed on the average anatomical brain image. **d** Peri-stimulus time histograms (psth) extracted from the ventromedial prefrontal cortex global maximum of activation in response to the stimulus value for each suggestion group with shadings corresponding to the standard errors of the mean. Light red – decreased hunger suggestion group, blue –increased hunger suggestion group. Note, the psth line graphs are an illustration of the activations shown in the SPM of (**c**). The coordinates correspond to the Montreal-Neurological-Institute (MNI) coordinates. Source data are provided as a Source Data file.

Fig. 3a, SI section 5). These findings suggest that the group difference between increased- and decreased-hunger suggestions on hunger ratings can be formally explained by the activation of interference-resolution regions located in the dlPFC at the time of decision.

## Suggestion-based placebo effects on the decision stage of economic choice

To gain more insight into whether and how suggestions affected the dynamics of the decision stage, choices and reaction times were fitted using a drift diffusion model that assumed that choice formation

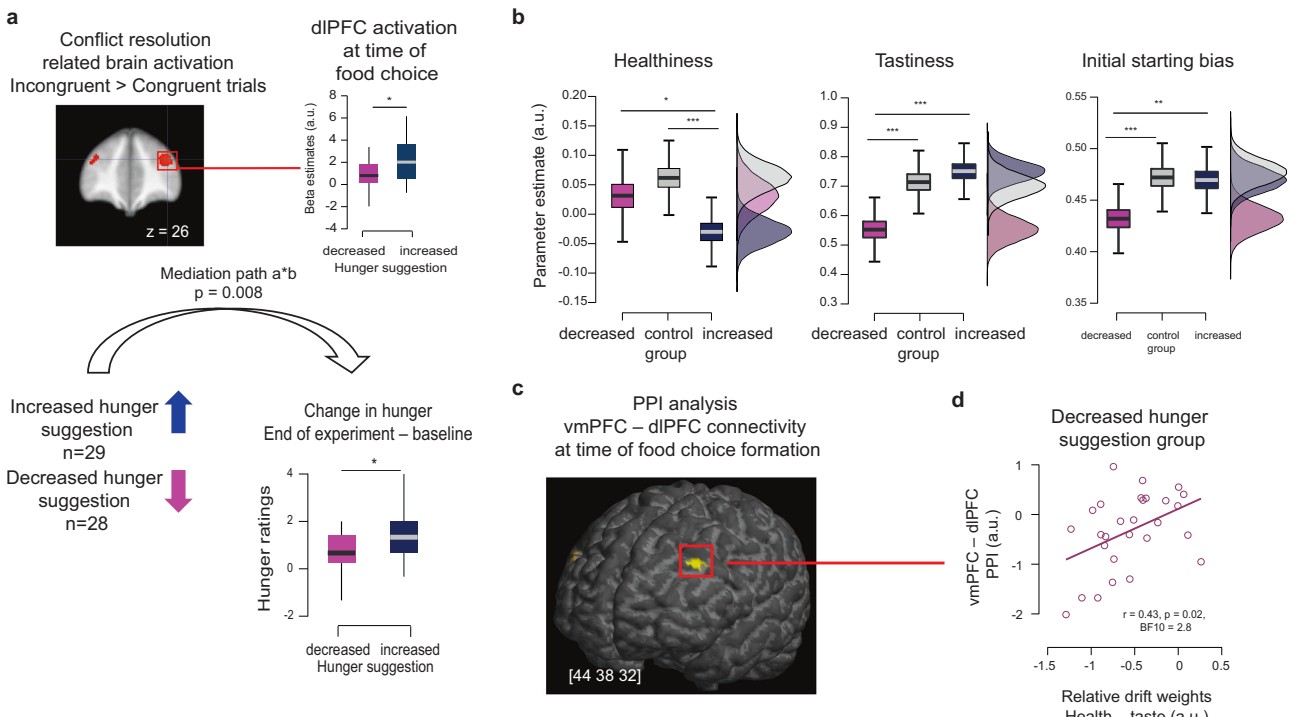

**Fig. 3 | Suggestion-based placebo effects on the decision stage of dietary decision-making. a** Brain mediation of suggestion-based placebo effectsRegion-of-interest (ROI)-based single-level mediation results for $N = 57$ participants of the fMRI experiment. SPMs display the significant voxels of the dlPFC ROI that were activated in response to interference resolution during the multi-source interference task (MSIT) at $p_{FWE} < 0.05$ corrected for multiple comparisons using family-wise error correction on the whole brain at the peak level. They are superimposed on the average anatomical image. Average path coefficients (a*b (SEM)) denote joint activation in paths a and b, and through it significant mediation. Boxplots display 95% confidence intervals for dlPFC MSIT ROI activation at time of food choice during the dietary decision-making task, and the change in hunger ratings from baseline to end of the experiment in decreased and increased hunger suggestion groups, respectively. Boxes correspond to the interquartile range from Q1 25th percentile to Q3 75th percentile. The gray and black lines indicate medians and whiskers range from minimum to maximum values and span 1.5 times the interquartile range. Horizontal lines indicate medians. **b** Individual parameters for tastiness, and healthiness drift weights and the starting point bias. The boxplots show 95% confidence intervals for the drift weights of healthiness, tastiness, and the initial starting point bias between the decreased-, control, and increased-hunger suggestion groups. The boxes of each boxplot show the interquartile range from Q1 25th percentile to Q3 75th percentile, horizontal lines indicate medians, the whiskers range from minimum to maximum values and span 1.5 times the interquartile range. ***$p_{mcmc} = 0.01$, **$p_{mcmc} = 0.02$, *$p_{mcmc} = 0.01$. The $p_{mcmc}$-values were obtained by comparing the proportion of posterior parameter differences from zero between groups. **c** Psycho-physiological interaction analysis. SPMs show significant voxels located in the dlPFC that interacted more strongly with the vmPFC seed ROI at the time of making the food choice. The yellow voxels are superimposed on a 3D anatomical brain image and survived small volume correction among the brain regions that were activated in response to interference resolution during the MSIT task. **d** Pearson's correlation between the vmPFC – dlPFC PPI and drift weight of healthiness relative to the drift weight of tastiness obtained in the decreased-hunger suggestion group ($N = 28$) from a separate 2 weight drift diffusion model (DDM). r –Pearson's correlation coefficient against zero, p values are two-tailed, BF Bayes factor. [x, y, z] coordinates correspond to the Montreal Neurological Institute space. Source data and exact p-values are provided as a Source Data file.

during the decision stage is the noisy accumulation of evidence in favor of a yes food choice over an alternative no food choice.

The drift rate of the noisy accumulation of evidence is influenced by a series of hidden latent parameters, such as the relative drift weights of healthiness and tastiness, the initial starting bias toward a yes or a no choice, or the sensorimotor integration to select a choice button (i.e., the non-decision time parameter).

Bayesian model comparisons showed that a standard 2 weight DDM, which assumed that the drift rate is influenced by both tastiness and healthiness evidence (Deviance Information Criterion (DIC) = 122,170) outperformed two alternative standard DDMs that either assumed the drift rate is solely influenced by tastiness (DIC = 123,299) or by healthiness information (DIC = 145,413).

Parameter comparisons between groups then showed that the strongest effects were observed for the tastiness drift weight. The values were overall, which indicated that tastier food was chosen more often, in line with similar results found by the GLM analyses of stimulus value (SI tables 2–8). Importantly, the influence of tastiness on the drift rate was stronger in participants of the increased hunger suggestion group compared to the decreased-hunger suggestion group (mean

$(D_{increased\ -\ decreased}) = 0.20$, PP = 0.99, $p_{mcmc} = 0.01$, Fig. 3b, SI table 13a). In participants of the control group the tastiness was stronger compared to the decreased hunger suggestion group (mean $(D_{control\ -\ decreased}) = 0.16$, PP = 0.99, $p_{mcmc} = 0.01$, Fig. 3b, SI table 13b), and non-different from the increased hunger suggestion group (SI table 13c).

In comparison to taste, healthiness had a relatively small influence on the drift rate, but still differed between the hunger suggestion groups. Healthiness weighted more positively on the drift rate in the decreased hunger suggestion group compared to the increased hunger suggestion group (mean $(D_{decreased\ -\ increased}) = 0.06$, PP = 0.95, $p_{mcmc} = 0.05$, SI table 13a). The control participants were non-different from the decreased hunger suggestion group (SI table 13b), but were significantly different compared to the increased hunger suggestion group (mean $(D_{controls\ -\ increased}) = 0.10$, PP = 0.99, $p_{mcmc} = 0.01$, SI table 13c).

Beyond the relative weights on taste and healthiness, a similar pattern was observed for the initial starting point bias in the DDM. The starting point bias was overall positive, and indicated an initial bias toward a yes food choice in all three groups. However, the yes initial

starting bias was weaker under the decreased hunger suggestion compared to the increased hunger suggestion- (mean ($D_{increased - decreased}$) = 0.04, PP = 0.99, $p_{mcmc}$ = 0.01, Fig. 3b, SI table 13a) and control groups (mean ($D_{control - decreased}$) = 0.05, PP = 0.99, $p_{mcmc}$ = 0.01, Fig. 3b, SI table 13b). The increased hunger suggestion and control groups did not differ, significantly (SI table 13c).

## Placebo effects on decision stage-related brain responses

To localize where in the brain the placebo intervention affected the decision stage, we searched for the psychophysiological interaction (PPI) of the vmPFC at the time of food choice. We focused on the dlPFC based on previous work that provided evidence for the implementation of action selection during the decision process by a vmPFC–dlPFC interaction[19,20,27,30,31,34]. Moreover, we found that a dlPFC ROI activation at time of food choice formation, and linked to interference resolution in an independent localizer task, mediated the suggestion effects on hunger (Fig. 3a). In accordance with the literature and our mediation results, the PPI analysis indicated significant covariance of the vmPFC with the dlPFC at the time of choice formation for all participants and groups ($p_{FWE}$ < 0.05 small volume corrected, Fig. 3c, SI Table 20 for whole brain activation). Average beta coefficients were extracted from this vmPFC – dlPFC PPI activation and then correlated with the relative drift weights of healthiness and tastiness. We observed a significant positive correlation ($R$ = 0.42, $p$ = 0.02, $BF_{10}$ = 3.1) for participants of the decreased-hunger suggestion group (Fig. 3d). Participants in the decreased-hunger suggestion group, who considered the healthiness more during evidence accumulation, were also those who displayed stronger vmPFC – dlPFC PPI connectivity during the decision stage of choice formation. The moderation of vmPFC – dlPFC connectivity by healthiness (relative to tastiness) was non-significant for participants in the increased-hunger suggestion group ($R$ = −0.008, $p$ > 0.05).

## Discussion

This study combined computational approaches with brain imaging and behavioral testing to provide insight into the putative mechanisms of suggestion-based placebo effects on appetitive interoceptive hunger experiences and hunger-addressing value-based decision-making. We used a validated placebo intervention that consisted of the administration of an inactive substance (i.e., a glass of water) together with the suggestion that the substance either increases or decreases hunger[15]. We then characterized the directionality of hunger suggestion effects on behavior and judgments by comparing them to the effects observed in a control group of participants who drank a glass of water without receiving information about its potential effects on hunger. In accordance with the results of a previous study[15], the intervention was successful in persuading participants about how efficiently the drink would decrease or increase hunger with an effect that was stronger in the two hunger-suggestion groups than the control group. Consistent with these stronger expectations and their framing, the decreased-hunger suggestion group reported feeling less hungry over the course of the experiment than the control group. On the contrary, the increased-hunger suggestion group reported to be hungrier than the control group. These findings indicate that the suggestion successfully induced prognostic beliefs about hunger and, through them, biased how participants sensed their hunger over the course of the experiment.

Importantly, we provide evidence for the underlying putative neurocognitive processes of such suggestion-based appetitive placebo effects. The strength of the prognostic belief in the effectiveness of the drink to decrease hunger moderated mPFC activation during food choice formation. Consistent with this finding, computational modeling further dissected this effect by showing that the suggestions about hunger influenced the valuation and decision stages of choice formation and the implementation of these two stages during economic choice formation by the vmPFC and dlPFC.

Past studies have reported mPFC activation in encoding[35] and computing belief-guided contextual reward expectations[36,37] or in representing lower pain expectations under placebo hypoalgesia[38]. Our results provide evidence for recruitment of the mPFC during decision-making as a function of participants' higher order beliefs about the effectiveness of a placebo drink to halt falling energy stores. The moderation was located within the anterior part of the dlPFC on the medial frontal gyrus, encompassing Brodmann's Area 10 and extending into the dorsal anterior cingulate cortex midway between the vmPFC and more posterior regions of the dlPFC, which have been shown to be part of the brain's valuation and cognitive regulation system that selects actions under the influence of self-control[19,20,30,31,39,40]. Our finding may thus suggest that participants in the context of a decreased-hunger suggestion made more self-controlled food choices. Indeed, we initially pre-registered this study under the hypothesis that cognitive regulation, such as self-control, would play a mediating role for the placebo effects on hunger. We also initially found evidence for a mediating role of regulatory success, which was defined by the propensity to make more restrained food choices (i.e., choose healthy, untasty food more often than tasty, unhealthy food, see pre-registered hypotheses).

However, dietary self-control is commonly measured by a participant's propensity to successfully stick to an a-priori set up goal, such as losing weight or healthier dieting, when being tempted by immediate tasty, unhealthy food reward. In this study, participants were asked to rate their natural food preferences under the suggestion that they had been administered a substance that was designed to either decrease or increase their hunger. The experimental design of our study was different from studies of dietary self-control[19,20,30,31,39]. The participants were not explicitly instructed to remember making choices to meet their personal goal under conflict. Moreover, we found that a slightly more posterior dlPFC region, which was linked to interference resolution, mediated the observed suggestion-based placebo effects. However, it mediated these effects in the opposite direction than what is known about the recruitment of more anterior dlPFC regions (including the mPFC) under dietary self-control[19,20,30,31,39]. The dlPFC interference resolution ROI activated more at time of food choice in the increased than decreased hunger suggestion group, and predicted positively the increase in hunger from baseline to the end of the experiment. Given both, the difference in our experimental design to dietary self−control studies and this mediation result, it cannot be fully inferred that participants in the decreased-hunger suggestion group were more self-controlled.

Following an information-theoretical approach to cognitive regulation[40], the placebo intervention may have generated other more contextual and/or perceptual forms of control. For example, it may have generated a form of contextual control that consisted in a modulation of value assigned to food during dietary decision-making. We directly tested this idea by building on generic models of economic choices and on evidence that the vmPFC is a central hub of the brain's valuation system that computes expected and experienced values across different decision-making problems and domains[23–26,41–44]. Our results converged on the finding that different contextual suggestions about hunger modulated hunger-addressing food choices and how the vmPFC computed the values assigned to food to guide these choices.

Another, mechanisms of cognitive regulation involved under the observed placebo effects consists in perceptual attentional filtering during the decision-making process[28]. Attentional filtering can be defined by reducing the cost of processing task-irrelevant information, such as overcoming interference during a Stroop effect[45]. During dietary decision-making under the influence of higher order prognostic beliefs about hunger, attentional filtering could consist of considering belief-relevant information more and neglecting belief-irrelevant information. For example, tastiness information might become more relevant for food choices in the increased hunger

suggestion group, and healthiness more relevant in the decreased hunger suggestion group.

Our computational modeling results provided some direct evidence for this idea. A drift diffusion model fitted to choices and reaction times disentangled several alternative hypotheses about how the placebo intervention shaped the decision stage of food choice formation. Direct evidence for considering belief relevant information is reflected by the result that the participants in the decreased-hunger suggestion group were less initially biased toward a yes food choice, and considered the healthiness more than participants in the increased-hunger suggestion group. On the contrary, the increased-hunger suggestion group was more initially biased toward yes food choices, and considered the tastiness more during choice formation.

These computational findings were underpinned at the neural level by moderation of the interaction between the vmPFC and dlPFC during the decision stage. The vmPFC has been reported to implement action selection in connection with other fronto-parietal brain regions, such as the dorsolateral prefrontal cortex[19,20,27,34]. That is why we used the vmPFC as a seed region of interest for searching for interactions with the dlPFC during food choice formation. We found that the more healthiness information weighed on the drift rate relative to tastiness, the more the vmPFC specifically interacted with the dlPFC during choice formation for participants of the decreased-hunger suggestion group. This study thus shows that placebo interventions generated higher order beliefs about hunger states and, potentially through them, shaped how extensively healthy participants weighed choice attributes and the brain implemented this biased action selection.

Most of the moderation results were specific to participants in the decreased-hunger suggestion group, which also showed differences in the behavioral and computational parameters from the control, no-suggestion group. The increased hunger suggestion group was in many findings non-different from the control, no-suggestion group. It is, therefore, possible that the decreased hunger suggestion group displayed a placebo effect, whereas the increased-hunger suggestion group reflected the natural fluctuation of hunger over the course of the experiment, similar to the control group. However, expectations and hunger ratings were significantly greater in the increased-hunger suggestion group than the control group. On the contrary, the health drift weight influenced the accumulation of evidence toward a food decision similarly in the control and the decreased hunger suggestion groups, and different to the increased hunger suggestion group. More research is therefore needed to characterize the increased-hunger suggestion group relative to the control group at the neural, hormonal, and cognitive levels.

Moreover, hunger and food choices were not fully independent. It may be that the beliefs instilled by the placebo intervention first changed hunger sensations and, subsequent to this change, the participant's propensity to pay attention to information during food choice formation that was more congruent (relevant) to their hunger sensation. More direct evidence is needed to fully disentangle the causal links between these nested effects. At the behavioral level interference resolution tasks unrelated to food choices, but sensitive to detect inter-individual differences could be used to test the causal links between attentional mechanisms, hunger sensations and food choices. Similarly, at the neural level, the use of brain imaging tools such as electroencephalograms can gain a better temporal resolution for detecting fast, perceptual attentional mechanisms at work under placebo conditions and early during the decision stage.

Interoception corresponds to the sensing of bodily states[46]. Here, we used a measurement of interoceptive sensitivity by asking participants to self-evaluate their hunger after self-reported fasting. The question is still open as to the extent hunger can be objectively sensed and whether a person's beliefs about her hunger can affect other dimensions of interoception, such as behavioral and neural interoceptive accuracy and metacognitive confidence in bodily signal

detection. A limitation of our results is also associated with potential biases induced by inter-individual differences in hormonal patterns and self-reported fasting. This study did not collect data on objective hormonal markers of hunger or information about the phase of the participant's menstrual cycle. Although similar suggestion-based placebo interventions have been demonstrated to affect ghrelin release[14,15], it is still unknown how hormonal variables would have affected hunger-addressing economic behavior and the effects of the suggestion-based placebo interventions. Future studies should explore potential hormonal effects on dietary decision-making under the influence of idiosyncratic higher order beliefs about the body.

Overall, our findings provide insights into the basic mechanisms of suggestion-based, appetitive placebo effects, and through them, contribute new knowledge to better understand placebo effects more broadly, notably, suggestion-based placebo effects on non-aversive domains of behavior and experiences beyond the pain and clinical domains. This is important for building a valid neurocognitive model of placebo effects across multiple domains. In the real world, it is irrelevant to provide false information about the ingredients of a substance to steer individuals towards a healthier diet or elicit a behavioral change. However, our findings provide the promise for the development of interventions to openly and non-deceptively target the cognitive processes of preference-based eating behavior. For example, they contribute to the neurobiological validation of communication-based interventions that act on higher-order beliefs about the body, reinforce personal reasons for behavioral change, and through them, might favor active treatment effects on altered hunger sensations and eating disorders.

## Methods

### Ethical considerations

The research in this study complied with all relevant ethical regulations. The study protocol followed the Declaration of Helsinki and was approved by the local ethics committee (Comité de Protection des Personnes Ile de France VI no 1204 and EST III no 19.12.05). All participants have provided written, informed consent.

### Study-pre-registration

The fMRI part of the study was initially pre-registered on the Open Science Framework before the start of data collection (https://osf.io/nw8c9).

The deviations from the pre-registered protocol involved:

(1) Our initial pre-registered sample size for the fMRI study of $n = 70$ was not possible to attain due to the challenges of in-person data collection during the COVID-19 pandemic. We recruited $n = 62$, which is a sufficiently representative sample to replicate a one-sided effect of suggestion on hunger (Cohen's $d = 0.6$) and follows recommendations for fMRI studies (see paragraph on sample size).

(2) A no-suggestion control group was added to better characterize the directionality of suggestion effects on the behavioral level.

(3) Moreover, we chose to more carefully position our results with respect to the initial hypothesis about the mediating role of cognitive regulation in terms of self-control, and provide rationales for this more careful positioning in the discussion.

### Participants

In total, 278 participants (mean age = $34.66 \pm 0.80$ years) were recruited for the study via a public advertisement in the Paris area. Of the 278 participants, 62 underwent functional magnetic resonance imaging.

All participants were screened for normal to corrected-to-normal vision, no history of substance abuse or any neurological or psychiatric disorders, and no medication. Participants of the fMRI experiment were additionally screened for the absence of metallic devices and

right-handedness. Participants were allotted to three experimental groups (see below), which were matched in age, level of education, and body composition (see SI table 15 for sociodemographic information).

All participants were tested in the morning between 8 am and 12 pm after overnight fasting (see SI table 16 and section 2.6 for healthy eating indexes (HEI) on the day before the experiment). Participants were asked to fast overnight and to not drink tea or coffee within least 2 h before arriving for the experiment.

Inclusion was restricted to self-identified female participants for the following reasons. Studies of placebo effects on pain have provided robust evidence that women display higher levels of positive expectations and greater placebo effects than men[47–49]. Importantly, we used a suggestion-based placebo intervention on hunger from a previous behavioral study of placebo effects on objective (i.e., hormonal) and subjective (i.e., self-reports) markers of hunger[15]. A secondary finding of this study was a significantly stronger placebo effect for women than men, which further motivated the recruitment of women for the current study. Empirical findings from another stream of research also provided convergent evidence that the physiology of appetite and eating behavior differs between men and women[50–54]. Women have been shown to display stronger brain activation to appetite-enhancing food stimuli, which they also liked more[52,53]. At the same time, women also show a stronger propensity for dietary self-restraint, which is potentially linked to sociocultural and psychological factors, such as perceived pressure to be thin[52]. To reduce these sources of variance, solely women were recruited for the current study.

Exclusion criteria were a baseline hunger rating <2 (no hunger), pregnancy, claustrophobia, permanent make-up or metallic implants that were not reported at the time of recruitment, and technical problems with the fMRI scanner. Based on these exclusion criteria, 23 participants were excluded from the data analysis due to problems with the fMRI scanner ($n = 1$) and not being hungry after overnight fasting at baseline ($n = 5$ in the decreased-hunger suggestion group, $n = 10$ in the increased-hunger suggestion group, and $n = 7$ in the control group).

Following the application of these exclusion criteria, 88 (28/88 with fMRI data) participants in the decreased-hunger suggestion group, 84 (29/88 with fMRI data) in the increased-hunger suggestion group, and 83 in the control, no-suggestion group were included in the analyses.

Participants were paid 15 euros for their participation in the behavioral experiment and 60 euros for their participation in the fMRI experiment.

## Sample size calculation

The sample size for the behavioral experiment was determined based on a moderate suggestion-based placebo intervention effect (Cohen's $d = 0.5$) on subjective hunger ratings reported by ref. 15. The software G*power (version 3.1) indicated that a sample between 102 and 118 participants ($n = 51–59$ within each hunger-suggestion group) was necessary to replicate a significant difference between the two extreme groups (enhanced satiety vs appetite suggestions) with a Cohen's $d = 0.5$, a statistical power between 80 and 85%, and a 5% chance of type I error (alpha = 0.05).

The sample size for the fMRI experiment was based on the findings from the prior behavior only experiment (Mean$_{decreased\ vs\ baseline}$ = $0.21 \pm 1.2$ < Mean$_{increased\ vs\ baseline}$ = $0.96 \pm 1.068$), which had an effect size of Cohen's $d = 0.6$. To replicate this one-sided effect of hunger suggestions (increased > decreased) on hunger ratings with a statistical power equal to 70% and a 5% risk for type I error, a minimum sample of 62 ($n = 31$ in each group) was necessary. We initially pre-registered the fMRI study with a planned sample size of 70 participants for the fMRI sample ($n = 35$ in each suggestion group) for a two-sided contrast of these two independent suggestion groups. Participants for the fMRI experiment were recruited between June 2020 and June 2021

in the midst of the COVID-19 pandemic, which was challenging for in-person testing, with frequent last-minute cancellations. We therefore stopped recruitment at 62 participants ($n = 31$ in each group). This sample size was a compromise between meeting the sample size recommendations for conducting task-based fMRI studies[54–56] and feasibility in terms of participant recruitment and the cost of fMRI.

Note that the suggestion-based placebo effects on hunger ratings were replicated in the smaller fMRI sample (see SI Fig. 1) and the effects were not significantly different in magnitude between the pilot and fMRI sample, as indicated by a non-significant hunger suggestion group$_{increased\ vs\ decreased}$ by testing time$_{baseline\ vs\ end\ of\ experiment}$ by sample $_{behavioral\ pilots\ vs\ fMRI}$ interaction: F(2,343) = 0.4; $p = 0.67$. We therefore pooled the two groups for the comparison of the decreased- and increased-hunger suggestion groups.

The sample size ($n = 90$) of the control group was defined to match the decreased- and increased-hunger suggestion groups combined.

## Randomization

The probability of being assigned to one of the two hunger suggestion arms was set to $p = 0.5$ and remained the same for the duration of the study. Randomization was performed before participants were enrolled using standard permutation algorithms implemented in MATLAB. The algorithm drew one of the two integers 1 and 2. If the integer was '1', the participant was assigned to the suggestion group 1 (decreased-hunger suggestion). If it was '2', the participant was assigned to suggestion group 2 (increased-hunger suggestion). To ensure an equal number of participants in each suggestion group the permutation was repeated 63 times for the behavioral pilots and 31 times for the fMRI participants.

Note, the initial pre-registration of the study did not foresee a control group, which was added a-posteriori. The randomization was therefore conducted on the two hunger suggestion groups.

## Placebo intervention

All participants were administered a glass of mineral water at the beginning of the experiment (®Eau minérale Evian Naturelle). The control group was told that the drink was just mineral water. For the two hunger-suggestion groups, the label on the water bottle was specifically designed to provide information about the water's ingredients to either decrease or increase hunger (Fig. 4). In addition, for the two suggestion groups, the experimenter explained the labels and all participants read an information booklet about the water's ingredients and their respective effects on hunger (for further details on the placebo intervention, see SI section 7).

Briefly, after rating their baseline hunger, the participants were assigned to the decreased- or increased-hunger suggestion group according to the randomization.

Participants in the increased-hunger suggestion group were told that the drink (water) was enriched with zinc, iron, and plant-based supplements, such as St. John's Wort, because these ingredients are known for their powerful stimulating effect on appetite through the potentiation of hunger-stimulating hormones, such as ghrelin. By contrast, participants in the decreased-hunger suggestion group were told that the water was enriched with vitamin B12, iron, and riboflavin, because these ingredients had a powerful effect on appetite to curb food cravings through the potentiation of hunger hormones, such as leptin.

The experimenters made sure that the participants understood the information about the drink before pouring it into a 25 ml glass (0.845 oz). Participants were then required to drink the whole glass within approximately 2–5 min. The experimenters were both female and male, and not blind to the placebo interventions. They left the room when participants of the behavioral experiment performed the dietary decision-making task.

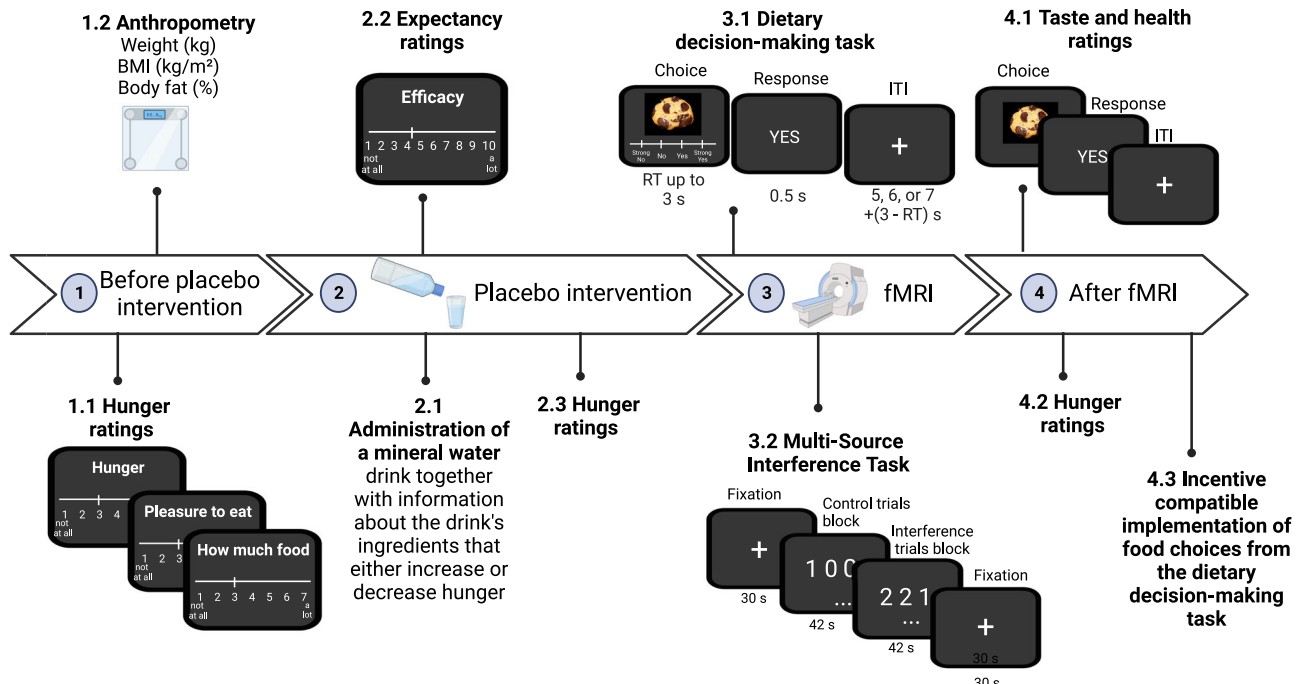

**Fig. 4 | Experimental procedure.** The scheme illustrates the temporal organization of the experiment with the different steps indicated by the numbers 1–4. The black panels correspond to computer screenshots of individual events for the hunger and expectancy ratings given before and after fMRI (outside the MRI scanner) and the dietary decision-making task and multi-source interference task performed during fMRI. The duration of the two fMRI tasks is shown in seconds. Created with Biorender.com.

## Expectancy ratings

After drinking the glass of water and before performing the dietary decision-making task, participants rated their expectation about how efficiently they believed the water drink would decrease or increase their hunger. Thus, they answered the following question: On a scale from 1 to 10, 1 being the minimum, how much do you expect this water to decrease (for the decreased-hunger suggestion group) / increase (for the increased-hunger suggestion group)/affect (control group) your hunger? All participants rated their expectations about the water drink.

## Hunger ratings

Hunger was assessed by three factors: (1) overall experienced hunger (i.e., How hungry do you feel?), (2) homeostatic hunger (i.e., How much food could you eat right now?), and (3) hedonic hunger (i.e., How pleasant would it be to eat, now?). Responses were given on a 7-point Likert scale from 1 (not at all) to 7 (very much) and averaged to a common score across the three questions. Hunger ratings were collected at two times during the behavioral study: (1) at baseline, before the placebo intervention, and (2) at the end of the experiment. They were collected three times during the fMRI experiment: (1) at baseline, (2) after the placebo intervention but before starting the fMRI session, and (3) at the end of the experiment.

## Dietary decision-making task

All participants performed a dietary decision-making task[19,20,30,31,39] and a sub-group of 61 performed the task during fMRI. The task consisted of the participants choosing whether they wanted to eat snack foods of varying tastiness and healthiness on a trial-by-trial basis. The task counted 200 trials (behavioral pilot) or 152 trials (fMRI sub-group) for a total duration of 20–30 min (Fig. 4). Each trial started with the display of a food item on a computer screen and participants indicated whether they wanted to eat the food item using a 4-point-Likert scale, from a strong no to a strong yes.

All food stimuli were selected from a database of 500 food images validated for tastiness and healthiness ratings by 300 participants from a non-published Mturk study conducted in-house (SI table 17). The food images were presented on a computer screen in the form of high-resolution images (72 dpi) as 100 g or 100 ml servings of the snack and its packaging following the protocol of past studies[39,57]. MATLAB and Psychophysics Toolbox extensions[58] were used for presentation of the stimulus and recording of the responses.

Participants of the fMRI experiment saw the stimuli via a head-coil-based mirror and indicated their responses using an fMRI compatible response box system.

After the dietary decision-making task, participants rated each food for its tastiness and healthiness using the same 4-point Likert scale used for the dietary decision-making task. These ratings were collected for all participants and outside the fMRI scanner.

The dietary decision-making task was incentive compatible, because one food was chosen by chance for consumption at the end of the experiment following similar approaches adopted by studies of value-based dietary decision-making[19,20,30,31,39]. In more detail, to ensure that participants' choices were close to their real preferences, they were told that at the end of the experiment, one of the food items would be randomly selected to count among all the items presented during the task. If the subject responded Yes or Strong Yes, they could eat that food before leaving the laboratory. Because only one random trial was selected to count, the optimal strategy was to treat each decision as if it were the only one. The foods that were eventually presented to the participant for consumption at the end of the experiment comprised chocolate bars, fruits (e.g., banana, clementine), or dairy-based drinks. Participants did not know what food they were going to be presented during the experiment and only found out at the very end before leaving the laboratory.

## MRI data acquisition

T2*-weighted multi-echoplanar images (mEPI) were acquired using a Siemens 3.0 Tesla VERIO MRI scanner with a 32-channel phased array

coil. Three echos were acquired for the best compromise between spatial resolution and signal quantity in the orbitofrontal cortex (OFC)[59,60]. To further reduce signal drop out in the OFC, we used an oblique acquisition orientation of 30° above the anterior–posterior commissure line[61]. This correction for signal dropout was applied because the ventromedial prefrontal cortex, a brain region of interest, encompasses the medial portions of the orbitofrontal cortex. Each volume comprised 48 axial slices collected in an interleaved manner. To cover the entire brain, the acquisition sequence involved the following parameters: echo times of 14.8, 33.4, and 52 ms; FOV = 192 mm; voxel size = 3 × 3 mm; slice thickness = 3 mm; flip angle = 68°; and TR = 1.25 s. Whole-brain high-resolution T1-weighted structural scans (1 × 1 × 1 mm) were acquired for all 61 subjects and co-registered with their mean mEPI images and averaged together to permit anatomical localization of the functional activation at the group level.

#### fMRI preprocessing

Image analysis was performed using SPM12 (Welcome Department of Imaging Neuroscience, Institute of Neurology, London, UK). Preprocessing involved the following steps: segmentation of the anatomical image into gray matter, white matter, and cerebrospinal fluid tissue using the SPM12 segmentation tool. The three echo images of each fMRI volume were summed into one EPI volume using the SPM12 Image Calculator[62–64]. Then, the summed EPIs were spatially realigned and motion corrected, co-registered to the mean image, and normalized to the Montreal Neurological Institute (MNI) space using the same transformation as for the anatomical image. All normalized images were spatially smoothed using a Gaussian kernel with a full-width-at-half-maximum of 8 mm.

#### Behavioral analyses

Statistical tests were conducted using the MATLAB Statistical Toolbox (MATLAB 2018b, MathWorks), R (3.3.2 GUI 1.68) within RStudio (RStudio 2022.02.3 + 492), and JASP (JASP 0.16.4). We conducted main effects tests using linear mixed effects (LME) models fitted to expectancy ratings, hunger ratings, and stimulus value ratings, controlling for BMI at the fixed effects level. Post-hoc two-tailed and one-tailed frequentist and Bayesian t-tests were used to further characterize the directionality of detected main effects and interactions, respectively.

#### Suggestion-based placebo intervention effects on expectancy ratings

To test whether the placebo intervention successfully induced expectations about the drink's effectiveness to increase or decrease hunger, a linear mixed effects model was fitted to expectancy ratings following Eq. 1:

$$\text{expectancy ratings} = \beta_0 \text{Intercept} + \beta_{\text{Group}}\text{Group} + \beta_{\text{BMI}}\text{BMI} + (\text{Intercept}|\text{Participant}).$$

(1)

This model included fixed effects for the intercept, suggestion group (coded −1 for decreased, 1 for increased, and 0 for control), BMI, and random intercepts nested by participant number.

#### Suggestion-based placebo effects on hunger ratings

For each session (baseline, end of experiment) hunger ratings were averaged for the three hunger questions to form one hunger score for each participant. A linear mixed effects model was fitted to these average hunger ratings following Eq. 2:

$$\text{hunger ratings} = \beta_0 \text{Intercept} + \beta_{\text{Group}}\text{Group} + \beta_{\text{Time}}\text{Time} + \beta_{\text{BMI}}\text{BMI} + \beta_{\text{Group} \times \text{Time}}\text{Group} \times \text{Time}(\text{Intercept}|\text{Participant}).$$

(2)

The model included the following fixed effects regressors: intercept, suggestion group (i.e., coded −1 for decreased hunger, 1 for increased hunger, and 0 for control), measurement time (i.e., coded −1 for baseline and 1 for the end of the experiment), BMI, and the main interest suggestion group by measurement time interaction. The model controlled for individual differences in hunger ratings by including a random intercept nested by participant number. Post-hoc paired and two-sample t-tests were conducted to characterize the main effects (of group, time) and interaction (group by time). Pearson correlations were performed for both hunger suggestion groups to assess how the hunger ratings were associated with the expectancy ratings.

#### Suggestion-based placebo effects on dietary decision-making

A linear mixed effects model was fitted to the mean centered stimulus value ratings from the dietary decision-making task following Eq. 3:

$$SV = \beta_0 \text{Intercept} + \beta_{\text{Group}}\text{Group} + \beta_{\text{BMI}}\text{BMI} + (\text{Intercept}|\text{Participant}).$$

(3)

The model included fixed effects for the intercept, the main effect of group (coded −1 for the decreased-hunger suggestion group, 1 for the increased-hunger suggestion group, and 0 for the control group) and BMI. It also controlled for individual differences in food valuation by a random intercept nested by participant number. Two sampled, two-tailed t-tests, along with Bayesian independent sample t-tests, were conducted to compare the average stimulus value ratings (SV) between groups.

Moreover, to also test how the hunger suggestions affected the computation of food preferences at the valuation stage, a multilevel general linear model (GLM) was fitted to stimulus value ratings (SV) following Eq. (4):

$$SV = \beta_0 + \beta_{\text{HR}}\text{HR} + \beta_{\text{TR}}\text{TR} + \beta_{\text{trial}}\text{trial} + \beta_{\text{HR*TR}}\text{HR*TR} + \beta_{\text{HR*trial}}\text{HR*trial} + \beta_{\text{TR*trial}}\text{TR*trial} + \varepsilon$$

(4)

At the individual level, the GLM assumed that the food SV was determined by the linear integration of tastiness (TR) and healthiness (HR) attributes of the food, with the rate of integration (beta weights, $\beta$) varying idiosyncratically between participants. This assumption is consistent with many other decision-making problems and at the core of the valuation phase proposed by models of economic choices. The GLM also included a trial number (trial) regressor to control for fatigue effects and three interactions (TR*HR, TR*trial, HR*trial) to assess how much change occurred in the weights given to the tastiness and healthiness attributes across trials and relative to each other. The SV, TR, and HR regressors were mean centered (i.e., coded −2 (strong no), −1 (no), 1 (yes), or 2 (strong yes)). Individual beta weights for each regressor (i.e., $\beta$) were then fitted into a second-level random effects analysis using two-tailed, two-sample t-tests to compare the two suggestion groups. More fine-grained analyses on dietary decision-making are reported in the Supplement (SI section 2 and SI tables 2–8).

To determine whether the placebo intervention also affected how calorie content moderated dietary decision-making, the calorie density (i.e., calories / 100 g or 100 ml serving) for each food item was provided by the food database, and used to divide the dietary decision-making trials into two trial categories, i.e., low-calorie and high-calorie food choices using a median split over the calorie densities for each participant (average median = 1.48, SEM = 0.13, range = 5.99). Pearson's correlations were then computed within each participant between calorie density and stimulus value for low and high calorie choice trials, respectively. Moreover, a linear mixed effects model was fit to stimulus

value ratings following Eq. 5:

$$SV \sim 1 + group + calorie\_level + group*calorie\_level + (1 \mid sub\_ID)) \quad (5)$$

The LME tested whether the stimulus value (SV) assigned to food was determined by the group (decrease: −1, control: 0, increase: −1) and/or the calorie level (low: −1 vs high: 1), and importantly, whether group and calorie level interacted on food valuation.

## Computational modeling

To test how and when suggestions about hunger influenced latent variables of the action selection (decision) stage of the decision-making process, SVs were collapsed into binary yes/no choices and fitted together with reaction times using a drift diffusion model (DDM), which has been validated by two independent studies for dietary decision-making[32,33].

Similar to traditional sequential sampling frameworks[65–67], the model assumed that committing to a choice results from the noisy accumulation of evidence up to a certain threshold in favor of one choice (say a yes) over an alternative choice (say a no). The DDM further assumed two sources of evidence: The tastiness (TD) and the healthiness (HD) of the food, which linearly scaled ($\omega_{tastecp}$, $\omega_{healthcp}$) the drift rate ($E_{cpt}(t)$) of evidence accumulation within the interval between the reaction time and the non-decision time (DT = RT - nDT). For example, the drift rate ($\delta_{cpt} = E_{cpt}(t)$) was determined at each timestep (t with dt = 8 ms) by Eq. 6[32]:

$$E_{cpt} = E_{cpt-1} + (\omega_{tastecp} * TD + \omega_{healthcp} * HD) * dt + noise \quad (6)$$

The differences in the tastiness and healthiness ratings for choosing a food item (yes response) versus not (no response) for given trials were denoted by TD and HD. Their respective weights on the updating of the evidence (the drift rate) were denoted by $\omega_{tastecp}$ and $\omega_{healthcp}$.

Overall, fitting the choices and reaction times with this 2 separate weight DDM allowed us to break down the action selection phase into the following hidden latent variables that were then compared between groups to test how they were influenced by the contextual hunger suggestion: (1) the respective influence of tastiness and healthiness on the evidence accumulation toward a yes over a no choice (e.g. drift rate weights), (2) the initial starting bias toward a yes or no food choice, and (3) the non-decision time, which approximated the time taken to initiate a choice and the corresponding motor response.

## Model specification

The model was specified using the RWiener package via the run.jags function of the JAGS package in RStudio. More specifically, a one-dimensional Wiener process implemented the DDM, where the state of evidence ($dE_t$) at each timestep (dt) evolved stochastically following differential Eq. (7):

$$dE_t/dt \sim N(E_t, \sigma^2) \quad (7)$$

$E_t$ was the evidence accumulation defined by Eq. 6 above. In practice, a stochastic node (y) reflected a certain state of evidence at a specific timestep (dt) (or the predicted choice data and reaction times) and was distributed according to a univariate Wiener distribution (8):

$$y_{cpt} \sim dwieners (\alpha_t = 2, \tau_{cpt}, \beta_{cpt}, \delta_{cpt}, \alpha_{cp}) \quad (8)$$

Choice and reaction time (RT) data were coded in a way that no food choices were given negative RT values and yes food choices positive RT values.

The evidence accumulation started with an initial value of evidence equal to the value of the starting bias parameter ($\beta_{cpt}$), which

was allowed to vary between participants as a random effect (more details about the priors for ß are provided in SI section 4.1). The boundary separation parameter ($\alpha_t$) was fixed to a maximum value of 2 on a trial-by-trial basis but varied between participants as a random effect. Since each participant was allowed to still have their own boundary separation parameter, the prior for the participant specific alpha ($\alpha_{cp}$) was drawn from a joint normal distribution: $\alpha_{cp} = N(\mu_{\alpha cp}, \sigma^2_{\alpha cp})$, with a mean $\mu_{\alpha cp,}$ that was itself drawn from a continuous uniform distribution between 0.001 and 2 and a variance $\sigma^2_{\alpha p}$ drawn from a gamma distribution with a shape of 1 and a rate of 0.1.

The model estimated the noise in the drift rate ($\delta_{cpt}$), which differed on a trial-by-trial basis and between participants. The prior for the drift rate was drawn from each trial (t) from a normal distribution: $\delta_{cpt} = N(E_{cpt}, e.p. \mathcal{T}_{cp})$, with a trial-specific mean that corresponded to the evidence ($E_{cpt}$) accumulated up to this trial following Eq. 6 and a variance (e.p. $\mathcal{T}_{cp}$) drawn from a gamma distribution with a shape and rate determined by the error terms of the regression function (see SI section 4.1) that was truncated between 0.001 and 2. The priors for the tastiness ($w_{tastecpt}$) and healthiness drift weights ($w_{healthcpt}$) were defined by uniform distributions between −5 and 5. Both drift weight-free parameters were allowed to vary between participants as random effects.

The non-decision time ($\tau_{cpt}$) was also allowed to differ between participants as a random effect, with a mean drawn from a uniform distribution between 0 and 10 and a variance drawn from a gamma distribution with a shape of 1 and a rate of 0.1 (see SI section 4).

## Model estimation

Groups were estimated, separately. The five free parameters of the model ($\alpha_p$, ß, $\omega_{taste,}$ $\omega_{health}$, $\tau$) were estimated by Gibbs sampling via the Markov Chain Monte-Carlo method (MCMC) in JAGS to generate posterior inferences for each parameter. A total of 5000 samples was drawn from an initial burn-in step and then three chains of 10,000 samples were run. Each chain was derived from three different random number generators with different seeds (see SI table 18). We applied a thinning of 10 to the final sample, which resulted in a final set of 30000 samples for each parameter. Gelman-Rubin tests were conducted for each parameter to test for the convergence of chains. The potential scale reduction factor (psrf) did not exceed 1.02 for any parameter at the participant or population level, and the deviance (the log posterior) had a prsf ~ 1. The autocorrelation between the chains was low in all three groups and for all five parameters (AC, 100-0.0).

## Model selection criteria

Choice and reaction time data were fitted using a standard two weight DDM, which was compared to two alternative DDMs with one weight, respectively: A health weight DDM, and a taste weight DDM. Deviance information criteria were used to compare the model fits. The DIC was defined following Gelman et al.[68] by Eq. 9:

$$DIC = 0.5^* \, var \, (deviance) + mean \, (deviance) \quad (9)$$

DICs were smaller for the 2 weight DDM (DIC = 12.2170) than the taste weight DDM (DIC = 12.3299) and the health weight DDM (DIC = 14.5413). Parameter recovery, reported in supplementary information (SI section 4.2), provided evidence that the parameter estimates were identifiable (SI table 19).

## Comparison of free parameters between groups

To determine whether latent, hidden parameters of the tDDM were different between groups, the posterior probability of potential group difference was calculated following Eq. 10.

$$PP = mean \, ((d_{increased} - d_{decreased}) > 0) \quad (10)$$

In more detail, a total of 3 posterior parameter distribution chains, counting each 10,000 samples, were concatenated for each parameter into one chain per parameter counting each 3,000 values (e.g., $PP_{increased}$, $PP_{decreased}$, $PP_{controls}$). For each parameter a difference between groups was calculated leading to a binary vector of the length 30,000. The values in this vector were coded 0 if the difference between groups was smaller then zero, and 1 if the difference was greater then zero (i.e., $D_{decreased\ vs.\ increased\ suggestion\ group}$, $D_{decreased\ vs.\ controls}$, $D_{increased\ vs.\ controls}$). The mean of those 30 000 binary outcomes for each parameter corresponds to the posterior probability (PP) that the population parameter distributions differed between groups.

Note, except for the healthiness drift weight, all comparisons were made with the prior prediction (H1) that the difference in the posterior parameter distributions were the following: $D_{increased\ -\ decreased\ suggestion\ group} > 0$; $D_{controls\ -\ decreased\ suggestion\ group} > 0$; $D_{increased\ -\ controls} > 0$.

For the healthiness drift weight the opposite was assumed decreased > controls > increased (see SI Table 13a, b, c for group-level mean posterior distributions, and posterior distributions of parameters in each group).

## Brain imaging analyses

fMRI data were analyzed using Statistical Parametrical Mapping (SPM12, Welcome Department of Imaging Neuroscience, Institute of Neurology, London, UK)[58]. Analogous to the behavioral analyses, we searched for suggestion effects on brain responses related to the valuation and action-selection phases of dietary decision-making.

fMRI timeseries were fitted using multilevel general linear models (GLMs). A first GLM (GLM1) included the following regressors at the first level: an onset regressor at the time of food image display (boxcar duration: reaction time) that was parametrically moderated by the stimulus value and an onset regressor for missed trials (boxcar duration: 3 s). Regressors of non-interest included six realignment parameters (x, y, z, roll, pitch, and yaw) to correct for head movement. Boxcar functions for each trial were convolved with the canonical hemodynamic response function. Individual contrast images for onset choice and the parametric modulator, the stimulus value, were then fitted into a second-level random effects analysis that used two-sample t-tests to localize brain voxels that were activated differently at the time of choice formation and in response to the stimulus value in the decreased-hunger suggestion group ($N = 28$) relative to that in the increased-hunger suggestion group ($N = 29$). Moreover, to test whether brain responses at the time of choice were moderated by expectations about hunger outcomes, expectancy ratings were added as second-level covariates of the choice onset regressor in a separate GLM (GLM2). GLM2 included the onset regressor at the time of choice, with a duration corresponding to the reaction time and a missed trials onset regressor of a boxcar duration of 3 s at the first level. At the second level, one-sample t-tests were used to test how much expected hunger modulated brain responses at the time of choice onset in each suggestion group.

Statistical thresholds for the second level random effects comparisons were corrected for multiple comparisons using family wise error corrections (FWE) on the cluster and voxel level ($p_{FWE} < 0.05$). For cluster wise correction for family wise errors a non-liberal initial uncorrected threshold of $p < 0.001$ was considered.

## Time courses

We extracted the activation time courses at the maxima of interest for all reported time course analyses. This was the vmPFC value-encoding region defined by MNI = [0, 52, −12] (SI table 3). The response time courses were estimated using a flexible basis set of finite impulse responses separated by one TR of 1.25 s.

## ROI-based mediation analysis

To test whether the activation of interference resolution-related brain regions during dietary decision-making mediated the effect of the placebo intervention on hunger experiences, beta estimates were extracted at the time of food choice onset (GLM1) from the following regions of interest: dACC = [−4, 10, 52], right insula = [32, 22, 4], and dlPFC [40, 42, 26] (SI Table 6). The activation in these regions survived small volume correction (SVC) using 10-mm spheres centered on the dACC [3, 10, 47], insula [32, 23, 10], dlPFC [47, 33, 27], which were reported by Bush et al. 2003[1] to be a robust and reliable neural correlate for interference resolution in humans.

Three ROI-based single-level mediation analyses were conducted, one per ROI, and corrected for multiple comparisons using a Bonferroni corrected $p$-value of p = 0.05/3 = 0.02 to infer significance. The initial X variable in each mediation model was the suggestion group (coded −1 for the decreased hunger group and 1 for the increased-hunger group). The outcome Y variable was the difference in the hunger ratings (Δhunger = end of experience – baseline), with positive differences indicating greater hunger at the end than at the beginning of the experiment. Path a of the regression for each mediation model tested for a linear effect of suggestion (increased > decreased hunger) on ROI activation at the time of food choice formation. Path b tested for the correlation between this brain activation and hunger by controlling for the effect of suggestion (path a). The path c regression tested for the direct effect of suggestion on hunger and c' for the total effect controlled for the effect of the mediator variable. Finally, mediation was tested by the product of the path a and path b regression coefficients using the formula: a * b = c − c' (see SI section 5 for more details on each path's coefficients).

Bootstrapping was performed to test the significance of each path coefficient[69,70]. This involved estimating the distribution of individual path coefficients by randomly sampling, by replacement, 10,000 observations from the matrix of [a, b, c, c', a * b] path coefficients. Two-tailed weighted $p$-values were calculated from the bootstrap confidence intervals.

## Psycho-physiological-interaction (PPI) analysis

PPI analysis aimed to localize the brain regions that exhibited choice formation-related functional connectivity within the brain and how such connectivity was linked to free parameters of the tDDM model in each suggestion group.

As the seed region, we chose the vmPFC region of interest that also correlated more strongly with food stimulus values in the increased-versus decreased-hunger suggestion group (MNI = [0, 52, −12], SI table 3).

Functional timeseries were fitted by a third GLM (GLM3), with three onset regressors at the first level: the time of fixation (duration = 0.5 s), choice (duration = reaction time), and missed trials (duration = 3 s). Realignment parameters were included as regressors of non-interest to control for head movement. We then extracted average BOLD activity timeseries from a 5-mm sphere centered around the vmPFC ROI (MNI coordinates = [0, 52, −12]) for the contrast choice versus fixation and estimated a fourth GLM (PPI-GLM4), which included a psychological regressor that modeled the choice formation as: reaction time - long boxcars at the time of food choice onset, the physiological regressor of the BOLD activity timeseries of the vmPFC seed region, and the interaction of the psychological and physiological regressors, which was the PPI regressor of interest. Individual betas for this PPI regressor were fitted into a second-level random effects analysis using one-sample t-tests (See whole brain PPI activations in SI table 14).

## Definition of regions of interest (ROI)

We used a theory-driven approach to determine regions of interest for (1) the mediation analysis and (2) to threshold second-level statistical parametric maps (SPMs) for PPI regressor-related brain activation.

Previous work has shown that the decision (action selection) stage of value-based decision-making is implemented by the vmPFC in connection with other prefrontal brain regions, such as the dorsolateral prefrontal cortex (dlPFC)[27–31]. Importantly, this work showed that the dlPFC plays a moderating role for value encoding within the vmPFC under cognitive regulation, such as when participants forgo short-term rewarding tasty foods in favor of more abstract and long-term rewarding healthy foods[19,20,31]. We reasoned that this type of cognitive regulatory success requires more attentional resources, similar to overcoming interference during a Stroop effect. We therefore defined the dlPFC functionally using an independent interference–resolution localizer task, the MSIT task[45] (see SI section 6). In more detail, to extract beta values of this region at time of food choice, and to make small-volume corrections of the PPI SPM, a 10-mm sphere was centered on the dlPFC MNI coordinates ([40, 42, 26]) that were more strongly activated in incongruent than congruent MSIT trials (see SI table 21). Note, the vmPFC–dlPFC PPI connectivity also survived small-volume correction when using an anatomically defined dlPFC ROI that comprised gray matter from Brodmann areas 9 and 46 built using the SPM wfupickatlas.

### Linking tDDM drift weights to vmPFC-dlPFC implementation of evidence accumulation

Average beta coefficients, which reflected the vmPFC–dlPFC interaction strength at the time of food choice, were then extracted from a very localized 5-mm radius sphere centered around the significant voxels located within the dlPFC activation. These beta coefficients were then correlated across participants to the difference between the healthiness and tastiness drift weights from the tDDM ($w_{healthiness} - w_{tastiness}$) using Pearson's correlations for both hunger-suggestion groups.

### Reporting summary

Further information on research design is available in the Nature Portfolio Reporting Summary linked to this article.

## Data availability

The source data file for conclusions of figures and tables has been deposited on figshare (https://doi.org/10.6084/m9.figshare.24088152) and is provided in the Supplementary Information/Source Data file. The raw fMRI and behavioral data files (MATLAB datasets) can be provided upon request by contacting the corresponding author pending scientific review and a completed material transfer agreement.

## Code availability

The code used for the dietary decision-making task, for model-free and model-based analyses is available on the OSF repository (https://osf.io/2fnmj/?view_only=9aa6dbee396540408579d47a9e72dc66).

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

## Acknowledgements

We thank Hilke Plassmann, Bassem Hassan, Mathias Pessiglione, Jean Daunizeau, and Douglas Lee for helpful comments. Thanks also to Zeynep Yoldas for helping with the data analysis, and Nathan Ferchaud for help with data collection. This work was supported by core funding from the Paris Brain Institute Foundation.

## Author contributions

L.S. designed the study with advice from K.M., S.F. and P.F.; I.K., B.R., S.F. and L.S. collected the data; I.K. and H.D. analyzed the data under the supervision of L.S. and advice and tools provided by T.A.H.; I.K. and L.S. wrote the first version of the manuscript and all authors contributed to the final text.

## Competing interests

The authors declare no competing interests.
