## [Peer Review File · Nature Communications]

Mapping expectancy-based appetitive placebo effects onto the brain in womenReviewer #1 (Remarks to the Author):

Key results:

The main finding that I took away from the paper is that hunger is modifiable by suggestion. The hypothesis was that suggestions about the appetite effects of water affect the experience of hunger, dietary decision-making, and cognitive regulation. 188 participants (all self-identified as female) were given water to drink and were randomized to one of 2 'suggestion' conditions: that water they were given to consume was (1) appetite enhancing or (2) appetite decreasing. fMRI was conducted during a dietary decision-making task in 61 of the 188 participants and a time-varying drift diffusion model was used to assess when and where in the brain the suggestion intervention affected dietary decision-making. They found that the suggestion intervention generated accordant hunger expectancies i.e., greater hunger expectancies in the group presented information that water was appetite enhancing; reduced hunger expectancies in the group presented information that water was appetite reducing. This expectancy also moderated activity in the medial prefrontal cortex during the food decision-making task. Participants who were in the group given the suggestion that the water they were given enhanced hunger (increased hunger group), valued food more highly and displayed stronger ventromedial prefrontal cortex (vmPFC) activation in response to food value than participants in the group given the suggestion that the water they were given, reduced hunger. Greater activity in the vmPFC, nucleus accumbens, posterior cingulate cortex, bilateral posterior insula, and precuneus was associated more strongly with food value in the increased-hunger suggestion group than the decreased-hunger suggestion group. Drift diffusion modeling of choice formation showed those in the increased-hunger suggestion group described the palatability of the food more strongly and rapidly, whereas those in the decreased-hunger suggestion group considered the healthiness of the food more strongly and rapidly during the decision stage. A psychophysiological interaction of vmPFC-dIPFC connectivity at the time of food choice was conducted showed that there was a significant association between the vmPFC and dIPFC during choice formation for the sample. For participants in the decreased-hunger suggestion group, there was a significant positive correlation showing a stronger association between the healthiness relative to tastiness of food with the greater vmPFC – dIPFC connectivity interaction. This relationship was non-significant for the increased-hunger suggestion group.

Significance

I think that this is a novel and interesting study to understand how the experience of hunger can be manipulated.

Concerns

Abstract

I thought the abstract was too general. I would detail the results more precisely e.g., state the tasks used, the number of subjects tested behaviorally and then scanned. In service of this, I thought some of the final paragraph of the introduction could be reemployed in the abstract to make the findings clearer and more precise.

Introduction

It may be that I do not study placebo effects but study the effects of hunger - so what I thought was the most interesting aspect of this paper is that expectancies regarding hunger can be influenced by suggestion. I found the focus on 'placebo' effects in the introduction distracting and would have found the paper easier to understand if the conditions were described as 'suggestions' rather than 'placebos'. This would have implications for the title, introduction, method description, abstract etc but I think would make the findings clearer.

Method

Overnight fasting without an inpatient or monitored overnight stay is best checked with a fasting blood sugar assessment. Assessment of food and alcohol consumption on the day prior to the experiment would be helpful. A limitation of the study is that one could not say that individuals had fasted prior to the experiment, other than relying upon self-report.

Another design concern was that there was no control group who did not receive any suggestion or a group that was told – this is just water. The authors recognize this design flaw in the Discussion. Only individuals self-identifying as female were included. Please justify this.

In the main paper, describe the Multi-Source Interference Task and in the figure show whether this was conducted before or after the dietary decision-making task.

In reviewing the supplement, it only became clear to me that the Multi-Source-Interference task (MSIT) task served a purpose in assessing if hunger suggestions affects expectancies via attentional mechanisms. This was not clear from the main text of the paper. If there is a way of

better integrating the findings of the MSIT task with the main text – this would help clarify how the findings were made and why attentional mechanisms were discussed in the Discussion.

In the supplement, there was also mention of a Stroop localizer – mention of this in Figure 5 and in the main text would make sense if the results were used.

How were the 61 subjects of the 188 who underwent the behavioral task – chosen for the imaging component? I do not think that this was clearly described.

To clarify for the Dietary decision-making task, I understand the one food was chosen by chance for consumption at the end of the experiment – what food was this? Did the participant know what food they were going to be presented and if so, at what point were they aware of this? Was this the food that was evaluated for tastiness and healthiness? Also some idea of what the food image stimuli were would be helpful e.g., high-fat, high-sweet food images? Whether these were balanced or matched in any way.

Was there any assessment of foods that participants liked or disliked? I am concerned that this may have had an influence on performance on the Dietary Decision-making Task.

Were there any measures given to assess individual differences in suggestibility? Looking at Figure 2, there appear to be individual differences in both the decreased hunger and increased hunger groups.

The supplement also details how examining responses to high and low calorie food images were also analyzed. While there is a passing mention of this in the main text, I would do more. Either explain some of the main results from these analyses, or more clearly describe how responses to high and low calorie food images were analyzed and that the details are provided in the supplement.

Results

In the results section or in the supplement, add a table to describe the socio-demographics and behavioral characteristics of participants e.g., age, race, ethnicity, education status, handedness, hunger – for the behavioral task and also for the imaging task. To check – in the Summary form, participants were described as undergraduates – the mean age was greater than what would be expected for undergraduates. Some explanation or description of the age range may help here. Detail the findings of the MTurk study in the supplement as this formed the foundation of the images chosen for the dietary decision-making task.

In the results - Expectancies ratings section – describe verbatim the question that participants were asked. Also describe the number of participants who completed the expectancies ratings – were these only the participants who were in the fMRI component or in all of the participants who completed the behavioral component.

Rationale for using the vmPFC seed ROI in the PPI analysis – possibly move this from the MATERIALS AND METHODS of the paper to the DISCUSSION. To check what was the vmPFC ROI seed for the PPI chosen (was it MNI coordinates = [0, 52, -12]; or MNI = [40, 42, 26])? I was confused by the description under the heading dIPFC ROI definition.

Detail the length of time to complete drinking the glass of water and quantity administered. Were there instances where participants did not complete drinking the water? This could be placed in the Supplement.

Denote in the results, the sample size for the completion of the behavioral Dietary decision-making task and the fMRI version of this task; multi-source interference task; and hunger ratings.

For some reason I could not access the OSF link. I would add information about the availability of data.

Some of the information in the report summary would be useful in the paper. For instance, add the information regarding how power was estimated in the paper; add if the experimenter was present or not in the room and the experimenter was not blind; as well as the dates in which the data was collected.

Discussion

Consideration in the Discussion of the real-world implications of the study. Giving false information about the appetite enhancing or not properties of a liquid has little real-world implications because both types of information are false. However, how would authors see the 'real-world' implications of their study.

Reviewer #2 (Remarks to the Author):

I very much enjoyed reading this manuscript!

In their study the authors investigate psychological, computational and neural underpinnings of expectation effects induced by a classical placebo intervention on appetitive interoceptive experiences and related behaviour. The effect of the placebo intervention on hunger is assessed at the behavioural, computational and neural level. This allows them to delineate temporal and neural aspects how taste and health information of visual food stimuli is integrated pointing towards a key role vmPFC and dlPFC interaction in this mechanism.

The topic is both scientifically and clinically highly relevant (although the authors are unusually reserved regarding the doubtless translational value of the study e.g. regarding the treatment of obesity etc.). The study provides a substantial advance in our understanding of the mechanisms that underly the effects of prognostic beliefs / expectations on health outcomes and health related decision making, which have only rarely been investigated in the field of hunger and food valuation.

Overall, the manuscript is very well written and the methods seem sound. The authors should address the points below to allow for a final assessment of the manuscript.

Major points:

I see no indication that the study has been pre-registered. Isn't this a prerequisite for a publication in Nature Communication?

Please provide a rationale (and power calculation) for the different sample sizes in the behavioural and the neuroimaging study. The substantially different sample sizes should be made more transparent in the manuscript, including the abstract.

It should also be made more obvious that the study has only included female participants. (see below for a follow-up remark).

The experimental design is lacking a control group that would allow for dissecting the mechanisms of positive and negative expectancy on hunger and food valuation.

In my view this is not a detrimental flaw as the two conditions investigated (positive and negative expectancy) still allow for the proof-of-concept that expectancy does have an effect. But again, this might be addressed in a more salient way.

Rating scales: The wording and dimension of the scales used to assess e.g. expectancy ratings (but also the other behavioural results such as valuation etc) should be understandable in the Results section without having to double-check with the methods. The same applies to all figure legends where the range of potential ratings should be indicated on the y-axis. (e.g. from text to text; 1-10, Likert scale, for Figure 1).

Why were different Likert scales (1-7; 1-10) used for the different ratings?.

Why were no physiological measures such as blood-glucose levels, leptin, ghrelin etc. obtained? This would have been the perfect study to obtain these valuable measures and relate these to the subjective ratings of hunger and food valuation.

Other comments:

Neuroimaging: why were different methods used to correct for multiple comparison?

E.g. cluster correction was used for results reported on page 4 and following, but peak voxel correction was used for results reported in the supplement.

Given that the definition of ROIs used for small-volume correction is always somewhat arbitrary when selecting coordinates from previous studies I strongly recommend to base SVC on anatomically defined regions. This also makes sure that the size of the region that SVC is based on

is realistic (a 10mm sphere is rather small for several of the regions investigated).

Minor:

Please check page 9, lines 255-258. Is there a "with" missing after "scaled"? The sentence is unclear.

Was the phase of the menstrual cycle assessed in the participants? And would this allow for exploratory analyses regarding hormonal influences on expectation effects on interception?

A figure visualizing the experimental protocol with the different time points and tasks would be helpful.

Reviewer #3 (Remarks to the Author):

The study investigated how prognostic beliefs about hunger affect appetitive interoceptive experiences and economic behavior by combining a placebo intervention on hunger with computational modeling and functional magnetic resonance imaging. The results suggest that prognostic beliefs about hunger shape hunger experiences, food valuation, and food-value encoding in the prefrontal cortex. The study also found that taste and health information integration influence the placebo effects on food choices. The findings provide insights into the neurocognitive mechanisms underlying how higher order prognostic beliefs shape non-aversive interoceptive sensitivity and decision-making. The topic is interesting and relevant, the paper is very well written, and the methods appear sound. I enjoyed reading the article. The large sample size is very impressive, especially for a human fMRI study. My enthusiasm is however damped due to critical methodological issues.

My main concern is the missing control group. Half of the participants were told that the water drink would increase hunger, whereas the other half was told the opposite. Hunger ratings increased in both groups - which was to be expected considering the time participants stayed without food prior to the experiments. Prognostic beliefs about hunger for each group were different from one (one-sample t-test), suggesting that each intervention had an effect different from "no expected efficiency". But, considering participants' generally increasing hungriness, couldn't comparable prognostic beliefs about hunger not also stem from individuals who received water without suggestions? There were also no between-group effects, questioning whether the two interventions really induced different beliefs about hunger. Although significant correlation analysis and model parameter comparison revealed distinct differences between both intervention groups, it remains questionable whether any effect in each of the two groups would be different from a group where water, but no suggestions have been given? Without such a control group, it is impossible to say in which way suggestions affected participants' beliefs.

Why did authors apply only tDDM and sDDM and not also other model variants? The modeling approach was adapted from Maier et al. (Ref. 30). Maier et al. applied tDDM and sDDM, but also four other variants (see Suppl. Tab. 1 in Ref. 30). They then used BIC, instead of DIC, to identify the best fitting model. In the paper under consideration, authors argue that tDDM has been validated by two independent studies (Ref. 30 & 52). However, model fit may vary across samples and designs. The finding that tDDM, as the more "context-sensitive" model, outperformed sDDM is also not surprising.

Was the data tested for normal distribution? When first reading the Introduction, I was expecting a main effect test applied to the entire sample followed by post-hoc tests for each sub-group (increase and decrease hunger suggestion group). In the Introduction, authors should better motivate their hypotheses. Although the total sample size is impressive, the effect sizes for many tests appear rather weak. Authors should provide sample and effect sizes for each test.

What was the rationale behind applying the cluster instead of the voxel level threshold to most contrast images (and the voxel-level to only some)? Applying the cluster level to functional SPMs is unconventional since it can lead to a loss of spatial resolution, introduce bias in the analysis, and

increase the likelihood of false positives or false negatives.

Why only female participants? I understand authors' argument of gender-related dietary self-restraint, but sex hormonal responses in women are known to interfere with the valuation of food pictures, which may vary over menstrual cycle. What about contraceptives and their potential influence? Many previous reward studies tested only male participants because of these known sex hormonal interactions in women.

I couldn't find any information about participants' body weight. Did authors assess the BMI? Have all participants been lean? Anorexia, overweightness, and any form of obesity would represent highly influential factors on task performance and fMRI signals. Was the BMI (and age) counterbalanced across groups?

Dear Reviewers,

Thank you for the detailed attention you gave our manuscript “Taste matters: Mapping expectancy-based appetitive placebo effects onto the brain.” (NCOMMS-23-06025)

We are grateful for the opportunity to provide a major revision of our manuscript. The reviews were very constructive and helpful. They guided us during the process, and we believe the manuscript is significantly improved.

The revision package includes the revised manuscript, supporting information, and point-by-point responses to the reviewer’s comments.

We have copied the editor and reviewer’s comments in bold black typeface and our responses in dark blue below for your convenience.

Sincerely,
The corresponding author on behalf of all co-authors

Reviewer #1 (Remarks to the Author):

Key results:

The main finding that I took away from the paper is that hunger is modifiable by suggestion. The hypothesis was that suggestions about the appetite effects of water affect the experience of hunger, dietary decision-making, and cognitive regulation. 188 participants (all self-identified as female) were given water to drink and were randomized to one of 2 ‘suggestion’ conditions: that water they were given to consume was (1) appetite enhancing or (2) appetite decreasing. fMRI was conducted during a dietary decision-making task in 61 of the 188 participants and a time-varying drift diffusion model was used to assess when and where in the brain the suggestion intervention affected dietary decision-making. They found that the suggestion intervention generated accordant hunger expectancies i.e., greater hunger expectancies in the group presented information that water was appetite enhancing; reduced hunger expectancies in the group presented information that water was appetite reducing. This expectancy also moderated activity in the medial prefrontal cortex during the food decision-making task. Participants who were in the group given the suggestion that the water they were given enhanced hunger (increased hunger group), valued food more highly and displayed stronger ventromedial prefrontal cortex (vmPFC) activation in response to food value than participants in the group given the suggestion that the water they were given, reduced hunger. Greater activity in the vmPFC, nucleus accumbens, posterior cingulate cortex, bilateral posterior insula, and precuneus was associated more strongly with food value in the increased-hunger suggestion group than the decreased-hunger suggestion group. Drift diffusion modeling of choice formation showed those in the increased-hunger suggestion group described the palatability of the food more strongly and rapidly, whereas those in the decreased-hunger suggestion group considered the healthiness of the food more strongly and rapidly during the decision stage. A psychophysiological interaction of vmPFC-dIPFC connectivity at the time of food choice was conducted showed that there was a significant association between the vmPFC and dIPFC during choice formation for the sample. For participants in the decreased-hunger suggestion group, there was a significant positive correlation showing a stronger association between the healthiness relative to tastiness of food with the greater vmPFC – dIPFC connectivity interaction. This relationship was non-significant for the increased-hunger suggestion group.

Significance

I think that this is a novel and interesting study to understand how the experience of hunger can be manipulated.

We thank the reviewer for the encouragement and detailed attention allotted to our manuscript and their comments, which allowed us to improve our manuscript. We have addressed all concerns point-by-point below. For your convenience, all comments are in bold black typeface, and our responses are in dark blue.

Concerns

Abstract

I thought the abstract was too general. I would detail the results more precisely e.g., state the tasks used, the number of subjects tested behaviorally and then scanned. In service of this, I thought some of the final paragraph of the introduction could be reemployed in the abstract to make the findings clearer and more precise.

Thank you for suggesting the use of the last paragraph of the instruction. To follow your advice, we have reformulated the abstract to within the word limit and added information about the sample size, the design of the study, and the tasks. We have also more directly linked the suggestion effects to placebo effects to address your next point below about the effects of suggestion on expectations and hunger sensations.

The abstract now reads:

“Suggestions about hunger can generate placebo effects on hunger experiences. But, the underlying neurocognitive mechanisms are unknown. Here, we show in 255 women that hunger expectancies, induced by suggestion-based placebo interventions, determined hunger sensations and economic food choices. Functional magnetic resonance imaging in a subgroup (n=57/255) provided evidence that the strength of expecting the placebo to decrease hunger moderated medial prefrontal cortex activation at the time of food choice and attenuated ventromedial prefrontal cortex (vmPFC) responses to food value. Dorsolateral prefrontal cortex activation linked to interference resolution formally mediated the suggestion-based placebo effects on hunger. A drift-diffusion model characterized this effect by showing that the hunger suggestions biased participants’ food choices and how much they weighted tastiness against the healthiness of food, which further moderated vmPFC–dlPFC psychophysiological interactions when participants expected decreased hunger. Thus, suggestion-induced beliefs about hunger shape hunger addressing economic choices through cognitive regulation of value computation within the prefrontal cortex.”

Introduction

It may be that I do not study placebo effects but study the effects of hunger - so what I thought was the most interesting aspect of this paper is that expectancies regarding hunger can be influenced by suggestion. I found the focus on ‘placebo’ effects in the introduction distracting and would have found the paper easier to understand if the conditions were described as ‘suggestions’ rather than ‘placebos’. This would have implications for the title, introduction, method description, abstract etc but I think would make the findings clearer.

We agree that the observed effects were engendered by verbal suggestion. That is why we used the term ‘suggestion’ throughout the paper and called our groups ‘suggestion groups’ instead of ‘placebo groups’.

Suggestion is a frequent experimental approach in combination with the administration of an inactive substance (a placebo) to experimentally induce placebo effects, but it is also an experimental factor for instilling other types of effects, such as hypnosis. To specifically link our suggestion effects to placebo effects, we also used the term placebo (e.g., see Kirsch I 1999 on suggestion-based placebo and hypnosis effects). By combining suggestion with placebo, we joined studies in the small but growing body of basic research on the neural, cognitive, and computational mechanisms of placebo effects. Placebo effects are very well known from clinical trials or studies using pain as a model. However, not much is known about the neural and cognitive mechanisms of the ‘appetitive’ aspect of placebo effects, notably, placebo effects that can be observed in healthy participants’ economic behavior (e.g., decision-making, effort allocation, learning) and associated non-painful interoceptive or exteroceptive sensations, such as hunger or pleasantness of the taste. This study aimed to fill this gap in the domain of hunger sensation and value-based dietary decision-making. Using the term placebo combined with the experimental method of inducing placebo effects by verbal suggestion is therefore crucial for delineating the scope and main message of our manuscript.

We thank the reviewer for helping us to position this work more clearly to reach readers interested in the aspects of the study that lay beyond basic research on the mechanisms of the placebo effect. We have reformulated our abstract and introduction (pages 2–3, lines 34–69) to more precisely state this specificity of our study and more clearly link the two aspects, suggestion and placebo, of our experimental design.

Method

Overnight fasting without an inpatient or monitored overnight stay is best checked with a fasting blood sugar assessment. Assessment of food and alcohol consumption on the day prior to the experiment would be helpful. A limitation of the study is that one could not say that individuals had fasted prior to the experiment, other than relying upon self-report.

Unfortunately, we could not collect biological samples, such as blood or saliva, the day before the experiment due to logistical and regulatory constraints. Solely asking participants to refrain from breakfast in the morning, testing all participants in the morning from 9 am to 12 pm, and using an incentive-compatible decision-making paradigm was a good experimental solution to ensure that (1) the sample was homogeneous concerning the duration of self-reported fasting and (2) it was possible for participants to come to the experiment in a fasted state. Importantly, we followed similar approaches used by other studies of the neurocognitive mechanisms of value-based dietary decision-making (Hare et al. 2009, 2011, 2014, Hutcherson et al. 2012, Harris et al. 2013, Schmidt et al. 2018, Koban et al. 2023).

By contrast, the assessment of alcohol consumption was part of a computerized 24-hour recall questionnaire that ascertained the complete dietary intake of the previous 24 hours for the control participants and those who underwent fMRI. This questionnaire involved obtaining a list of all food items eaten on the previous day by a trained dietician (the 2nd author of the study). The interviewer probed for frequently forgotten foods, such as savory snacks, sugary beverages, candies, etc. Then, the time and name of the eating occasion (i.e., breakfast, lunch, dinner) and a detailed description of the type of food item, cooking method, and other relevant aspects were collected. This information was then used to calculate a healthy eating and energy intake index (HEI) from the day before for each participant.

In more detail, the HEI scores were computed based on the Dietary Guidelines for Americans 2015-2020 (DGA, and following Krebs and Smith 2008). These guidelines consist of 13 components scored following adequacy- or moderation-based energy-adjusted intake cut-offs, (except for the fatty-acid components). Each component weights differently in the final scoring, which ranges between 0-100, where higher scores indicate a healthier dietary pattern. The components and their respective

weights were: total fruits (5 points), whole fruits (5 points), total vegetables (5 points), greens and beans (5 points), whole grains (10 points), dairy (10 points), total protein foods (5 points), fatty acids (polyunsaturated fatty acid plus the monounsaturated fatty acid to saturated fatty acid ratio) (10 points), refined grains (10 points), sodium (10 points), added sugars (10 points), and saturated fats (10 points). Refined grains, sodium, added sugars, and saturated fats, as components to be consumed in moderation, were reversely scored (Krebs-Smith et al., 2018). A recall was also conducted one month after the experiment and the results were essentially the same, which indicates that this index is a stable measure of everyday diet across a period of one month.

We now report these HEI scores for each suggestion group and a control, no-suggestion group in novel SI table 10, which we show below for your convenience.

SI table 10. HEI scores for N = 57 participants of the fMRI experiment and N=54 participants of the control study

	Hunger suggestion	Mean	SEM	Minimum	Maximum
Day before	Decreased	48	2.9	20	78
	Increased	45	3.1	17	80
One month later	Controls	49	1.5	34	80
	Decreased	52	3.8	21	80
	Increased	51	2.8	16	81

Note, healthy eating indexes scores were calculated following Krebs-Smith et al. (2018). A score of 100 indicates total correspondence of eating behavior with the 2015-2020 Dietary Guidelines for Americans. No differences between HEI-2015 were observed for eating on the day before the fMRI, and one month later. SEM – standard error of the mean.

In summary, the table indicates that eating was comparable and no different between the two hunger-suggestion groups and the control group, and healthy eating patterns were still at similar levels one month after the experiment for the two suggestion groups. Although this evidence provides an indirect indication of the status of the energy stores on the day of the experiment, it provides some evidence that the eating habits of the participants did not differ the day before the experiment.

We agree with the reviewer that the lack of a direct objective measure of fasting is a limitation. We have now added this point to the discussion alongside a call for more research to determine other measurements of interoception beyond subjective hunger sensitivity and self-reported fasting (lines 518–526), which now read:

“A limitation of our results is also associated with potential biases induced by inter-individual differences in hormonal patterns and self-reported fasting. This study did not collect data on objective hormonal markers of hunger or information about the phase of the participant’s menstrual cycle. Although similar suggestion-based placebo interventions have been demonstrated to affect ghrelin release^{14,15}, it is still unknown how hormonal variables would have affected hunger-addressing economic behavior and the effects of the suggestion-based placebo interventions. Future studies should explore potential hormonal effects on dietary decision-making under the influence of idiosyncratic higher order beliefs about the body.”

Another design concern was that there was no control group who did not receive any suggestion or a group that was told – this is just water. The authors recognize this design flaw in the Discussion.

We initially pre-registered the study without a control group because past studies have shown that the results for a no-suggestion control group in these types of placebo interventions was located between those of the decreased- and increased-hunger suggestion groups. To address the open

question about the underlying cognitive, computational, and neural mechanisms, we initially focused on the two hunger-suggestion groups. As reviewer 2 pointed out, this is not a detrimental flaw, as our goal was to provide a proof-of concept that expecting an effect of water on hunger influences hunger-addressing economic choices, their underlying stages (valuation and decision stage), and how the brain computes value.

However, we also agree that a no-suggestion control group can potentially add more fine-grained information, in contrast to the two hunger-suggestion groups. We therefore re-collected behavioral data (decision making task, hunger ratings, 24-hour dietary recall, and questionnaires) on 83 participants matched for demographic and body composition to the hunger suggestion groups.

In summary, we found that the control group expected a significantly smaller effect of the water drink on their hunger. The suggestion-based placebo intervention instilled stronger expectations about the drink's efficiency to decrease/increase hunger in the two suggestion groups. Consistent with expectations, hunger increased more in the control group than in the decreased-hunger suggestion group and less than in the increased-hunger suggestion group.

These findings further validated our experimental scope to test how suggestions about decreased or increased hunger influence behavioral and computational parameters of hunger-addressing economic choices. Again, the results of the control group were located between those of the decreased- and increased-hunger-suggestion groups. We found linear main effects of group (coded - 1 for decreased, 0 for control, 1 for increased) for all these measurements, with the control group though often close to the increased hunger suggestion group (but see effects on the health drift weight from the DDM).

We have revised the results and figures accordingly and added a paragraph to the discussion (page 18, lines 489–501) to aid in the interpretation of these results, which reads:

“Most of the moderation results were specific to participants in the decreased-hunger suggestion group, which also showed differences in the behavioral and computational parameters from the control, no-suggestion group. The increased hunger suggestion group was in many findings non-different from the control, no-suggestion group. It is, therefore, possible that the decreased hunger suggestion group displayed a placebo effect, whereas the increased-hunger suggestion group reflected the natural fluctuation of hunger over the course of the experiment, similar to the control group. However, expectations and hunger ratings were significantly greater in the increased-hunger suggestion group than the control group. On the contrary, the health drift weight influenced the accumulation of evidence toward a food decision similarly in the control and the decreased hunger suggestion groups, and different to the increased hunger suggestion group. More research is therefore needed to characterize the increased-hunger suggestion group relative to the control group at the neural, hormonal, and cognitive levels.”

Only individuals self-identifying as female were included. Please justify this.

This study was conducted on female participants to keep the group homogeneous in terms of known sex/gender differences in placebo effects, dietary self-control, and eating behavior.

We have added a more complete rationale for this decision to the methods section page 21, lines 578–591, which now reads:

“Inclusion was restricted to self-identified female participants for the following reasons. Studies of placebo effects on pain have provided robust evidence that women display higher levels of positive

expectations and greater placebo effects than men⁴⁷⁻⁴⁹. Importantly, we used a suggestion-based placebo intervention on hunger from a previous behavioral study of placebo effects on objective (i.e., hormonal) and subjective (i.e., self-reports) markers of hunger¹⁵. A secondary finding of this study was a significantly stronger placebo effect for women than men, which further motivated the recruitment of women for the current study. Empirical findings from another stream of research also provided convergent evidence that the physiology of appetite and eating behavior differs between men and women⁵⁰⁻⁵⁴. Women have been shown to display stronger brain activation to appetite-enhancing food stimuli, which they also liked more⁵²⁻⁵³. At the same time, women also show a stronger propensity for dietary self-restraint, which is potentially linked to sociocultural and psychological factors, such as perceived pressure to be thin⁵². To reduce these sources of variance, solely women were recruited for the current study.”

In the main paper, describe the Multi-Source Interference Task and in the figure show whether this was conducted before or after the dietary decision-making task.

The MSIT was conducted after the dietary decision-making task to localize brain regions associated with cognitive regulation, such as the allocation of more attentional resources during interference resolution trials. This is now better highlighted (i.e., by numbering) in figure 4, which displays the various assessments during the experiment in chronological order.

Figure 4. Experimental procedure. The scheme illustrates the temporal organization of the experiment with the different steps indicated by the numbers 1 to 4. The black panels correspond to computer screenshots of individual events for the hunger and expectancy ratings given before and after fMRI (outside the MRI scanner) and the dietary decision-making task and multi-source interference task performed during fMRI. The duration of the two fMRI tasks is shown in seconds.

In reviewing the supplement, it only became clear to me that the Multi-Source-Interference task (MSIT) task served a purpose in assessing if hunger suggestions affects expectancies via attentional mechanisms. This was not clear from the main text of the paper. If there is a way of better integrating the findings of the MSIT task with the main text – this would help clarify how the findings were made and why attentional mechanisms were discussed in the Discussion.

We thank the reviewer for suggesting to better integrate the MSIT results in the main manuscript. Indeed, our pre-registered research question was about the mediating role of cognitive regulation processes, such as attentional mechanisms for expectancy-based placebo effects on hunger and hunger-associated dietary decision-making. The MSIT served as an independent localizer task to harness brain activation linked to attentional filtering during interference resolution. We found evidence that after correcting for multiple comparisons, a region of interest located in the posterior dIPFC fully mediated the placebo effect on hunger ratings. We have decided to move these results back into the main text on page 11, lines 282 – 301 and Figure 3a. This addition of the main results allowed us to more precisely delineate and discuss our findings in terms of attentional mechanisms beyond the initially pre-registered hypothesis of self-control. We now also discuss the meaning of these additional findings on page 16, lines 433–449, which read:

“However, dietary self-control is commonly measured by a participant’s propensity to successfully stick to an a-priori set up goal, such as losing weight or healthier dieting, when being tempted by immediate tasty, unhealthy food reward. In this study, participants were asked to rate their natural food preferences under the suggestion that they had been administered a substance that was designed to either decrease or increase their hunger. We further reason that our experimental design was different from studies of dietary self-control^{19,20,30,31,39}. The participants were not explicitly instructed to remember making choices to meet their personal goal under conflict. Moreover, we found that a slightly more posterior dIPFC region, which was linked to interference resolution, mediated the observed suggestion-based placebo effects. However, it mediated these effects in the opposite direction than what is known about the recruitment of more anterior dIPFC regions (including the mPFC) under dietary self-control^{19,20,30,31,39}. The dIPFC interference resolution ROI activated more at time of food choice in the increased than decreased hunger suggestion group, and predicted positively the increase in hunger from baseline to the end of the experiment. Given both, the difference in our experimental design to dietary self-control studies and this mediation result, it cannot be fully inferred that participants in the decreased-hunger suggestion group were more self-controlled.”

In the supplement, there was also mention of a Stroop localizer – mention of this in Figure 5 and in the main text would make sense if the results were used.

We apologize for being unclear. The Stroop localizer task was the MSIT task. It was used to localize brain regions that are associated with attentional resource allocation during interference resolution. We have checked the whole manuscript, SI, and figures to consistently replace Stroop localizer with MSIT.

How were the 61 subjects of the 188 who underwent the behavioral task – chosen for the imaging component? I do not think that this was clearly described.

Thank you for pointing out this need to clarify the explanation for the sample sizes and experimental groups.

Following standard methodological approaches for planned fMRI studies, we first tested participants behaviorally. This was done in an attempt to ensure that the experimental design led to behavioral results that then justified the use of fMRI to test how the brain underpins these effects. We therefore recruited 126 participants for the behavioral pilot experiment and then 62 for a pre-registered fMRI experiment.

The sample size for the behavioral pilot experiment was determined to replicate previously reported suggestion-induced placebo effects on subjective hunger ratings (see Hoffmann et al. 2018). This previous study recruited 90 participants assigned to three groups (e.g., control, enhanced appetite

suggestion, enhanced satiety suggestion), each group consisting of 30 participants. Hunger ratings were indicated on a visual 10 cm analogous scale. The mean hunger score in the enhanced-satiety suggestion groups was 4.3 cm (SD = 3.2) and that in the enhanced-appetite suggestion group 6.3 cm (SD = 2.3). Using the difference between the two extreme groups (enhanced satiety vs appetite), we calculated an estimated sample size to replicate an effect size of Cohen's $d = 0.5$ with a power between 80 and 85% and a 5% chance for type I error ($\alpha = 0.05$), which ranged between 102 and 118 participants ($n = 51$ to 59 within each group).

The sample size for the fMRI experiment was based on the findings from this prior behavior only experiment ($\text{Mean}_{\text{decreased vs baseline}} = 0.21 \pm 1.2 < \text{Mean}_{\text{increased vs baseline}} = 0.96 \pm 1.1$), which showed an effect size of Cohen's $d = 0.6$. To replicate this one-sided effect size, with a statistical power equal to 70% and a 5% risk for type I error, a minimum sample of 62 ($n = 31$ in each group) was necessary. We initially pre-registered the fMRI study with a planned sample size of 70 participants for the fMRI sample ($n = 35$ in each suggestion group) for a two-sided contrast of these two independent suggestion groups. Participants for the fMRI experiment were recruited between June 2020 and June 2021 in the midst of the COVID-19 pandemic, which was challenging for in-person testing, with frequent last-minute cancellations. We therefore stopped recruitment at 62 participants. This sample size was also a compromise between meeting the sample size recommendations for conducting task-based fMRI studies⁵³⁻⁵⁵ and feasibility in terms of participant recruitment and the costs of fMRI.

To address concerns raised by the reviewers, we recruited an additional sample of 90 control participants, with 83 included in the analyses after application of the same exclusion criteria as those applied to the suggestion groups. This sample size for the control group was chosen to match the joined behavioral and fMRI sample sizes for the decreased- ($n = 84$) and increased- ($n = 88$) hunger-suggestion groups.

Moreover, although not directly addressing the reviewer's comment, we show in the figure below that the suggestion-based placebo effects on hunger ratings were replicated in the smaller fMRI sample. In more detail, for the behavioral pilots, a significantly greater increase in hunger from baseline to the end of the experiment was observed for the increased-hunger (mean hunger increase_{baseline to end} = 0.96, sem = 1.1) than the decreased-hunger (mean hunger decrease_{baseline to end} = 0.22, sem = 0.14) suggestion group ($t(113) = 3.49$, $p < 0.001$, Cohen's $d = 0.61$). We observed a similar difference for the smaller fMRI sample: participants in the increased-hunger suggestion group (mean hunger increase_{baseline to end} = 1.35, sem = 0.22) reported being significantly hungrier at the end of the experiment than at baseline and than participants in the decreased-hunger (mean hunger decrease_{baseline to end} = 0.62, sem = 0.27) suggestion group ($t(55) = 2.11$, $p = 0.04$, Cohen's $d = 0.56$). The suggestion effect was not significantly different in magnitude between the pilot and fMRI samples, as indicated by a non-significant hunger suggestion group_{increased vs decreased} by testing time_{baseline vs end of experiment} by sample_{behavioral pilots vs fMRI} interaction: $F(2,343) = 0.4$; $p = 0.67$).

Figure S1. Placebo effects on hunger ratings for the pilot and fMRI samples. Raincloud plots for hunger ratings from baseline to the end of the experiment for the pilot behavioral sample (left side) and the fMRI sample (right side). Each dot corresponds to the difference in a participant’s hunger rating from baseline to the end of the experiment, with positive values indicating an increase in hunger ratings. Distributions of the increase in hunger ratings are shown to the right of each plot. Boxplots show the 95% CI for both hunger-suggestion groups and samples. * $p < 0.05$ for significant within-group differences.

We have added this more detailed information about sample size calculations to the methods section, page 22, lines 606–634, and present the figure showing the replication of the placebo effect on hunger sensation in section 1 of the supplement.

To clarify for the Dietary decision-making task, I understand the one food was chosen by chance for consumption at the end of the experiment – what food was this?

Foods were proposed to render the dietary decision-making task incentive compatible following standards in the field of value-based dietary decision-making. These foods involved snacks, such as chocolate bars, fruits (e.g., banana, clementine), and yogurt drinks.

Did the participant know what food they were going to be presented and if so, at what point were they aware of this? Was this the food that was evaluated for tastiness and healthiness?

During the dietary decision-making task, participants did not know what food they could eat at the end of the experiment. It was revealed at the end of the experiment before they left the laboratory. We now detail the incentive-compatible procedure in the corresponding methods section on page 25, lines 708 – 719, which reads:

“The dietary decision-making task was incentive compatible, because one food was chosen by chance for consumption at the end of the experiment following similar approaches adopted by studies of value-based dietary decision-making^{19,20,30,31,39}. In more detail, to ensure that participants’ choices were close to their real preferences, they were told that at the end of the experiment, one of the food items would be randomly selected to count among all the items presented during the task. If the subject responded “Yes” or “Strong Yes”, they could eat that food before leaving the laboratory. Because only one random trial was selected to count, the optimal strategy was to treat

each decision as if it were the only one. The foods that were eventually presented to the participant for consumption at the end of the experiment comprised chocolate bars, fruits (e.g., banana, clementine), or dairy-based drinks. Participants did not know what food they were going to be presented during the experiment and only found out at the very end before leaving the laboratory.”

Also some idea of what the food image stimuli were would be helpful e.g., high-fat, high-sweet food images? Whether these were balanced or matched in any way.

The food images were selected based on ratings from an unpublished food image database. This database consists of 500 snack food images that were rated by 300 participants online using Mturk (for more detail on this database study, see our response to a related comment below) on tastiness, healthiness, familiarity and liking using 100 mm visual analogous scale.

To answer the reviewer’s question more specifically, we added table 11 to the supplement that shows the average tastiness, healthiness, familiarity, and liking of the food stimuli used during the dietary decision-making task, alongside their average caloric, fat, sugar, and salt content and retail price. Moreover, these food stimuli were selected using the ratings from the database so that half were a-priori conflicting (i.e., 25% tasty & unhealthy, 25% untasty & healthy) and the other half non-conflicting (i.e., 25% tasty & healthy, 25% untasty & unhealthy).

SI table 11. Food stimuli used during the dietary decision-making task

Food stimulus category		Tastiness	Healthiness	Familiarity	Liking	Calories (kcal)	Fat (g)	Sugar (g)	Salt (g)	Total sugars (inc. sugar)	Price per 100 g/mL
conflict food images	tasty & unhealthy	5.4	3.0	3.7	4.3	384	20	31	1	42	1.49
	untasty & healthy	3.7	4.8	3.1	3.4	154	6	7	1	23	1.96
non-conflict food images	tasty & healthy	5.0	5.1	4.0	4.6	105	2	12	0	18	1.21
	untasty & unhealthy	2.7	2.6	2.7	2.7	248	13	9	2	22	0.96

Was there any assessment of foods that participants liked or disliked? I am concerned that this may have had an influence on performance on the Dietary Decision-making Task.

We used a-priori liking ratings from the food database to build the dietary decision-making food stimuli list, and to make sure liked and not liked food stimuli were distributed equally as a function of tastiness and healthiness (SI table 11, above). In a nutshell, the SI table 11 above shows that tasty foods are slightly most liked, and even more if they are also healthy. Our stimulus value ratings corroborate these a-priori liking ratings (see SI Figure S3 and SI section 2.4) with more frequent ‘yes’ choices for tasty & healthy and tasty & unhealthy foods, and less frequent ‘yes’ choices for less liked untasty & healthy and untasty & unhealthy foods.

We would like to argue that economic choices are assessed at the behavioral level by measuring the stimulus value assigned to food. In this study, the stimulus value was framed as a rating on how much participants actually wanted to eat the different food items at the end of the experiment. At this higher-cognitive level, the stimulus value ratings approximated food preferences in terms of both wanting and liking the food.

Pioneer studies in the field of neuroeconomics have used dietary decision-making as a model for investigating how the brain implements economic choices and, in particular, computes the valuation step of economic choices. These studies have provided convergent evidence that the stimulus value (i.e. preference) is computed by the ventromedial prefrontal cortex, irrespective of how its

behavioral measure was framed: as a binary food choice (Maier et al. 2020; Sullivan et al. 2021; Koban et al. 2023), wanting (e.g., Hare et al. 2009, 2011), willingness to pay (e.g., Hutcherson et al. 2012, Chib et al. 2009), or a liking rating (e.g., Harris et al. 2011). Moreover, the same brain regions have been shown to be consistently activated for stimulus values across these different value-rating modalities for food and non-food items (Chib et al. 2009: stimulus value framed as willingness to pay, Lebreton et al. 2009: stimulus value framed as a liking rating). These prior findings provide robust evidence for a modality-general valuation system in the brain (see also Bartra et al. 2013 for meta-analysis). Building on this literature, we therefore argue that neural and behavioral results are potentially the same if participants would have rated the food items on how much they liked them.

Moreover, economic stimulus value ratings are different from the judgement of hedonic experiences of food, such as hedonic 'liking', which can best be assessed by licking, pupil size, and facial expressions. The other 'hedonic' behavioral component of reward is hedonic 'wanting' or incentive salience, which is best measured by effort allocation or learning rates (e.g., Berridge and Robinson 2003). These hedonic components of reward are indeed dissociable at the neural and psychological level (for recent reviews see Morales and Berridge 2020). However, assessing how they influence economic choices and its core measure, the stimulus value, is beyond the scope of this study.

Were there any measures given to assess individual differences in suggestibility? Looking at Figure 2, there appear to be individual differences in both the decreased hunger and increased hunger groups.

Unfortunately, we did not assess suggestibility. However, we used the observed individual differences in each suggestion group to identify, for example, their association with psychological (e.g., the strength of expected drink efficiency on hunger), computational (i.e., relative drift weights and starting bias from a DDM of food choices and reaction times), and neural sources (e.g., mPFC activation at the time of food choice).

We found that these individual psychological, computational, and neural differences were specifically associated with individual differences in the expectations about the potential hunger-decreasing effect of water and decreased hunger ratings.

The supplement also details how examining responses to high and low-calorie food images were also analyzed. While there is a passing mention of this in the main text, I would do more. Either explain some of the main results from these analyses, or more clearly describe how responses to high and low calorie food images were analyzed and that the details are provided in the supplement.

We have added more detail about these analyses to the main text's methods section (page 29, lines 810–822), and explain now more precisely that estimates about the calorie content were obtained from the 500-food image database. These estimates in the database were calculated for a 100 g or 100 ml serving that was shown for each food on the corresponding image. We then split the food images into high and low calorie foods by using the median of these calorie estimates, and fitted a linear mixed effects model to stimulus value ratings to test for a linear main effect of group (decreased < control < increased) and calorie level (low < high) and an interaction group by calorie level.

Note, these analyses are complementary to the main analyses and results presented in the main manuscript. In other words, they show the same result obtained by applying a different perspective. We have changed how they are introduced in the main text results section on pages 8–9, lines 230–251 and Figure 2b, to be clearer about these complementary results for the main message and robustness of our findings.

Results

In the results section or in the supplement, add a table to describe the socio-demographics and behavioral characteristics of participants e.g., age, race, ethnicity, education status, handedness, hunger – for the behavioral task and also for the imaging task.

The socio-demographic information was shown in the supplement (SI table 9) and was referenced in the main text. In more detail, the table contains information about the age and level of education, along with information about body composition and the average questionnaire scores for eating disorders (6 question of the SCOFF, a score of 2 or more indicates suspicion of eating disorder), dysphoria (Beck's Depression Inventory, BDI), food addiction (Yale Food Addiction Scale, YFAS), and physical activity (in MET minute scores – metabolic equivalents per minutes performed, IPAQ-sf). For your convenience we show it below.

SI Table 9. Sociodemographic, body composition, and questionnaire data.

	Decreased (n=88)	Control (n=83)	Increased (n=84)
Age (years)	35.7 ± 1.4	32.3 ± 1.3	32.7 ± 1.4
Education	3.8 ± 0.1	3.7 ± 0.1	3.9 ± 0.04
Weight (kg)	63.7 ± 1.6	60.7 ± 1.3	63.3 ± 1.4
BMI (kg/m ²)	23.4 ± 0.5	22.2 ± 0.4	23.1 ± 0.4
Body Fat (%)	28.9 ± 1.1	28.1 ± 2.5	27.9 ± 1.1
SCOFF	1.4 ± 0.3	1.2 ± 0.1	1.3 ± 0.1
BDI	4.5 ± 0.5	2.6 ± 0.3	3.8 ± 0.5
YFAS	0.8 ± 0.2	1.1 ± 0.2	0.9 ± 0.3
IPAQ-sf	1626 ± 182	1988 ± 150	1360 ± 196

Note, education was coded as 1 for undergraduate, 2 for high school graduate, 3 for two years of higher education, and 4 for more than two years of university education. SCOFF, YFAS, and IPAQ average scores were calculated for the fMRI study participants only (N = 57). SCOFF – Sick, Control, One stone, Fat, Food; BDI – Beck's depression inventory, YFAS – Yale food addiction scale, IPAQ-sf – International Physical Activity Questionnaire short form.

Please note that this table does not contain information about ethnicity or race of the participants, because the processing of personal data revealing “racial” or ethnic origin is prohibited by French law (Data Protection Act No. 78-17, Article 6 created by the CNIL 1978 and in action since 2019).

Concerning laterality, this information was not collected for the behavioral pilot experiment participants. However, all participants in the fMRI experiment were right-handed following standard inclusion criteria for fMRI studies.

To check – in the Summary form, participants were described as undergraduates – the mean age was greater than what would be expected for undergraduates. Some explanation or description of the age range may help here.

We apologize for the confusion stemming from the use of the term “undergraduates” in the reporting summary and explanatory note of SI table 9. The average level of education was 3.8 and 3.9 in the hunger suggestion groups, and 3.7 for the control group. This corresponds to two and more than two years of higher academic education, which in France corresponds to a bachelor's degree level. The confusion stems from the fact that we initially took level 4 (more than 2 years of

college) as the threshold for defining college graduation, which corresponds to having graduated with a Master's or PhD degree. To avoid confusion we have removed the term "undergraduates" in the reporting summary, and clarified in the note of SI table 9 that a score of 1 indicates high school undergraduates, 2 high school graduates, 3 two years of higher education (college level) and 4 more than 2 years of than two years of college.

Detail the findings of the MTurk study in the supplement as this formed the foundation of the images chosen for the dietary decision-making task.

The Mturk study was conducted in-house by other colleagues and aimed to build an open-access 500-food image database. Please contact Dr. Hilke Plassmann for more details.

Briefly, the database is not yet published and sociodemographic data were not collected from the 300 participants, who rated each of the 500 food images based on tastiness, healthiness, liking, and familiarity on a 100-mm visual analogous scale. Moreover, the 500-food image database also included information about calories (kcal), fat (g), sugar (g), salt (g), carbohydrates (g incl sugar), and cost (€) per 100 g or ml servings. This information was obtained in-house by searching the packaging details provided by the manufacturers and retailers. The images themselves were presented as 100 g or ml servings of the snack food item and its packaging, or in case of fruit, the fruit as a whole, and a 100 g serving of it.

The 200 food stimuli selected from this 500-food image database were the same as those selected for a similar dietary decision-making task reported in Schmidt et al. 2018; and Koban et al. 2023.

We have added this information to the supplement in the form of SI table 11 and refer in the methods (page 25, lines 697– 702) that the food stimuli were used for similar dietary decision-making tasks in past publications (refs 39 and 57).

In the results - Expectancies ratings section – describe verbatim the question that participants were asked. Also describe the number of participants who completed the expectancies ratings – were these only the participants who were in the fMRI component or in all of the participants who completed the behavioral component.

The wording of the expectancy ratings was the following:

“On a scale from 1 to 10, 1 being the minimum, how much do you expect this water to decrease (for the decreased-hunger suggestion group) / increase (for the increased-hunger suggestion group) / affect (control group) your hunger?”

We have added this verbatim to the methods section of the main manuscript, page 24 lines 674–677 and corresponding figure legends.

The response to this question was collected for all participants.

Rationale for using the vmPFC seed ROI in the PPI analysis – possibly move this from the MATERIALS AND METHODS of the paper to the DISCUSSION.

To follow the reviewer's advice, we have moved the rationale for using the vmPFC as a seed ROI for the PPI analyses to the discussion, page 17 lines 478–488.

To check what was the vmPFC ROI seed for the PPI chosen (was it MNI coordinates = [0, 52, -12]; or MNI = [40, 42, 26])?

The seed ROI for the PPI analysis was located within the vmPFC defined by MNI coordinates = [0, 52, -12]. We have now clarified this on page 35, lines 990–992, which now read:

“As the seed region, we chose the vmPFC region of interest that also correlated more strongly with food stimulus values in the increased- versus decreased-hunger suggestion group (MNI = [0, 52, -12], SI table 3).”

The MNI coordinates [40, 42, 26] were located within the dlPFC, which was activated more strongly during incongruent trials in the independent MSIT localizer task. This functionally defined ROI was used (1) as a brain mediator for suggestion-based placebo effects on hunger and (2) for small-volume correction of the PPI SPMs following a theory-driven approach explained in more detail in response to your comment just below, and added to the methods section page 36, lines 1005–1023.

I was confused by the description under the heading dlPFC ROI definition.

We apologize for this unclear paragraph.

Our rationale for searching for a vmPFC–PPI within the dlPFC was the following. Adopting a theory-driven approach, we propose that the decision stage of value-based decision-making requires a weighting of information, which requires high attentional effort. Past studies have shown that this decision (action selection) stage is underpinned by stronger vmPFC–dlPFC functional (PPI) connectivity (Hare et al. 2009, 2011, 2014, Hutcherson et al. 2012, Domenech et al. 2017, Rudolf et al. 2014, Harris et al. 2013). In particular, the dlPFC has been shown to moderate value-encoding within the vmPFC when participants successfully overcome choice conflict between a short-term rewarding tasty food and a more abstract and sustainable healthy food choice. We reasoned that this type of cognitive regulatory success requires more attentional resources, similar to overcoming interference during a Stroop effect. We therefore localized interference-resolution related dlPFC activation using an independent MSIT and then used a 10-mm sphere centered on the MNI coordinates from the peak activation located in the dlPFC for incongruent compared to congruent MSIT trials for small volume correction of the PPI SPMs.

We have removed the paragraph defining the dlPFC ROI and have integrated a reformulation to better explain our rationale for using the dlPFC as a region of interest to threshold the vmPFC–PPI connectivity on page 36, lines 1005–1023. This part now reads:

“Definition of regions of interest (ROI)

We used a theory-driven approach to determine regions of interest for (1) the mediation analysis and (2) to threshold second-level statistical parametric maps (SPMs) for PPI regressor-related brain activation. Previous work has shown that the decision (action selection) stage of value-based decision-making is implemented by the vmPFC in connection with other prefrontal brain regions, such as the dorsolateral prefrontal cortex (dlPFC)²⁷⁻³¹. Importantly, this work showed that the dlPFC plays a moderating role for value encoding within the vmPFC under cognitive regulation, such as when participants forgo short-term rewarding tasty foods in favor of more abstract and long-term rewarding healthy foods^{19,20,31}. We reasoned that this type of cognitive regulatory success requires more attentional resources, similar to overcoming interference during a Stroop effect. We therefore defined the dlPFC functionally using an independent interference–resolution localizer task, the MSIT (see SI section 6). In more detail, to extract beta values of this region at time of food choice, and to make small-volume corrections of the PPI SPM, a 10-mm sphere was centered on the dlPFC MNI coordinates ([40, 42, 26]) that were more strongly activated in incongruent than congruent MSIT trials (see SI table 15). Note,

the vmPFC–dIPFC PPI connectivity also survived small-volume correction when using an anatomically defined dIPFC ROI that comprised grey matter from Brodmann areas 9 and 46 built using the SPM wfupickatlas.”

Detail the length of time to complete drinking the glass of water and quantity administered.

Were there instances where participants did not complete drinking the water? This could be placed in the Supplement.

All participants drank a 25 ml cup of water and were instructed to drink the whole cup within no more than 5 minutes. We have added this detail in the methods section, page 24, lines 666–668, which read:

“The experimenter made sure that the participants understood the information about the drink before pouring it into a 25 ml glass (0.845 oz). Participants were then required to drink the whole glass within approximately 2 to 5 min.”

Denote in the results, the sample size for the completion of the behavioral Dietary decision-making task and the fMRI version of this task; multi-source interference task; and hunger ratings.

For some reason I could not access the OSF link. I would add information about the availability of data.

We have added information about the sample sizes to the participants section of the methods, pages 20–21, lines 565–598.

Behavioral data is now accessible on the study’s Open Science Framework page (<https://osf.io/nw8c9>).

Some of the information in the report summary would be useful in the paper. For instance, add the information regarding how power was estimated in the paper; add if the experimenter was present or not in the room and the experimenter was not blind; as well as the dates in which the data was collected.

Thank you for this advice. All of this information has now been added to the corresponding methods sections and reporting summary.

The experimenter was not blind to the placebo interventions but left the room when participants of the behavioral experiment performed the dietary decision-making task. We added this information to the methods section about the placebo intervention (page 24, lines 668–670, which read:

“The experimenters were both female and male, and not blind to the placebo interventions. They left the room when participants of the behavioral experiment performed the dietary decision-making task.”

Discussion

Consideration in the Discussion of the real-world implications of the study. Giving false information about the appetite enhancing or not properties of a liquid has little real-world implications because both types of information are false. However, how would authors see the

'real-world' implications of their study.

Thank you for encouraging us to consider real-world applications of our basic research findings. We have followed this advice and similar advice from reviewer 2 and have added the following reasoning to the end of the discussion (page 19, lines 527–539):

“Overall, our findings provide insights into the basic mechanisms of suggestion-based, ‘appetitive’ placebo effects, and through them, contribute new knowledge to better understand placebo effects more broadly, notably, suggestion-based placebo effects on non-aversive domains of behavior and experiences beyond the pain and clinical domains. This is important for building a valid neurocognitive model of placebo effects across multiple domains. In the real world, it is irrelevant to provide false information about ingredients of a substance to steer individuals towards a healthier diet or elicit a behavioral change. However, our findings provide the promise for the development of interventions to openly and non-deceptively target the cognitive processes of preference-based eating behavior. For example, they contribute to the neurobiological validation of communication-based interventions that act on higher order beliefs about the body, reinforce personal reasons for behavioral change, and through them, might favor active treatment effects on altered hunger sensations and eating disorders.”

Reviewer #2 (Remarks to the Author):

I very much enjoyed reading this manuscript!

In their study the authors investigate psychological, computational and neural underpinnings of expectation effects induced by a classical placebo intervention on appetitive interoceptive experiences and related behaviour. The effect of the placebo intervention on hunger is assessed at the behavioural, computational and neural level. This allows them to delineate temporal and neural aspects how taste and health information of visual food stimuli is integrated pointing towards a key role vmPFC and dlPFC interaction in this mechanism.

The topic is both scientifically and clinically highly relevant (although the authors are unusually reserved regarding the doubtless translational value of the study e.g. regarding the treatment of obesity etc.).

We thank the reviewer for this encouraging feedback.

To be more explicit about the real-world application of our work, and also in response to a similar comment of reviewer 1, we have now added the following lines to page 19, lines 527–539, to the discussion:

“Overall, our findings provide insights into the basic mechanisms of suggestion-based, ‘appetitive’ placebo effects, and through them, contribute new knowledge to better understand placebo effects more broadly, notably, suggestion-based placebo effects on non-aversive domains of behavior and experiences beyond the pain and clinical domains. This is important for building a valid neurocognitive model of placebo effects across multiple domains. In the real world, it is irrelevant to provide false information about ingredients of a substance to steer individuals towards a healthier diet or elicit a behavioral change. However, our findings provide the promise for the development of interventions to openly and non-deceptively target the cognitive processes of preference-based eating behavior. For example, they contribute to the neurobiological validation of communication-based interventions that act on higher order beliefs about the body, reinforce personal reasons for behavioral change, and through them, might favor active treatment effects on altered hunger sensations and eating disorders.”

The study provides a substantial advance in our understanding of the mechanisms that underly the effects of prognostic beliefs / expectations on health outcomes and health related decision making, which have only rarely been investigated in the field of hunger and food valuation.

Overall, the manuscript is very well written and the methods seem sound. The authors should address the points below to allow for a final assessment of the manuscript.

Please find below our point-by-point responses.

Major points:

I see no indication that the study has been pre-registered. Isn't this a prerequisite for a publication in Nature Communication?

The fMRI replication study was preregistered on the OSF website and can be consulted using this link: <https://osf.io/nw8c9>

However, our protocol and the positioning of the results deviated from the initial pre-registration for several points. We have therefore added the following paragraph to the methods section page 20, lines 551–564:

“Study-pre-registration:

The fMRI part of the study was initially pre-registered on the Open Science Framework before the start of data collection (<https://osf.io/nw8c9>). The deviations from the pre-registered protocol involved:

- (1) Our initial pre-registered sample size for the fMRI study of $n = 70$ was not possible to attain due to the challenges of in-person data collection during the COVID-19 pandemic. We recruited $n = 62$, which is a sufficiently representative sample to replicate a one-sided effect of suggestion on hunger (Cohen's $d=0.6$) and follows recommendations for fMRI studies (see paragraph on sample size).
- (2) A no-suggestion control group was added to better characterize the directionality of suggestion effects on the behavioral level.
- (3) Moreover, we chose to more carefully position our results with respect to the initial hypothesis about the mediating role of cognitive regulation in terms of self-control, and provide a rationale for this more careful positioning in the discussion.”

Please provide a rationale (and power calculation) for the different sample sizes in the behavioural and the neuroimaging study. The substantially different sample sizes should be made more transparent in the manuscript, including the abstract.

Thank you for this advice. We have added more precise details on the sample sizes for fMRI and behavioral experiments in the abstract, and the calculations to the methods section on page 22, lines 606–634. The methods part now reads:

“Sample size calculation

The sample size for the behavioral experiment was determined based on a moderate suggestion-based placebo intervention effect (Cohen's $d = 0.5$) on subjective hunger ratings reported by Hoffmann et al. 2018. The software G*power (version 3.1) indicated that a sample between 102 and 118 participants ($n = 51$ to 59 within each hunger-suggestion group) was necessary to replicate a significant difference between the two extreme groups (enhanced satiety vs appetite suggestions)

with a Cohen's $d = 0.5$, a statistical power between 80 and 85%, and a 5% chance of type I error ($\alpha = 0.05$).

The sample size for the fMRI experiment was based on the findings from the prior behavior only experiment ($\text{Mean}_{\text{decreased vs baseline}} = 0.21 \pm 1.2 < \text{Mean}_{\text{increased vs baseline}} = 0.96 \pm 1.1$), which had an effect size of Cohen's $d = 0.6$. To replicate this one-sided effect of hunger suggestions (increased > decreased) on hunger ratings with a statistical power equal to 70% and a 5% risk for type I error, a minimum sample of 62 ($n = 31$ in each group) was necessary. We initially pre-registered the fMRI study with a planned sample size of 70 participants for the fMRI sample ($n = 35$ in each suggestion group) for a two-sided contrast of these two independent suggestion groups. Participants for the fMRI experiment were recruited between June 2020 and June 2021 in the midst of the COVID-19 pandemic, which was challenging for in-person testing, with frequent last-minute cancellations. We therefore stopped recruitment at 62 participants ($n = 31$ in each group). This sample size was a compromise between meeting the sample size recommendations for conducting task-based fMRI studies⁵³⁻⁵⁵ and feasibility in terms of participant recruitment and the cost of fMRI.

Note that the suggestion-based placebo effects on hunger ratings were replicated in the smaller fMRI sample (see SI Figure 1) and the effects were not significantly different in magnitude between the pilot and fMRI sample, as indicated by a non-significant hunger suggestion group_{increased vs decreased} by testing time_{baseline vs end of experiment} by sample_{behavioral pilots vs fMRI} interaction: $F(2,343) = 0.4$; $p = 0.67$). We therefore pooled the two groups for the comparison of the decreased- and increased-hunger suggestion groups.

The sample size ($n = 90$) of the control group was defined to match the decreased- and increased-hunger suggestion groups combined."

It should also be made more obvious that the study has only included female participants. (see below for a follow-up remark).

We thank the reviewer for steering us to be clearer about this point. We have added a more detailed rationale for including only women in this study to the methods section, page 21, lines 578–591, which now reads:

"Inclusion was restricted to self-identified female participants for the following reasons. Studies of placebo effects on pain have provided robust evidence that women display higher levels of positive expectations and greater placebo effects than men⁴⁷⁻⁴⁹. Importantly, we used a suggestion-based placebo intervention on hunger from a previous behavioral study of placebo effects on objective (i.e., hormonal) and subjective (i.e., self-reports) markers of hunger¹⁵. A secondary finding of this study was a significantly stronger placebo effect for women than men, which further motivated the recruitment of women for the current study. Empirical findings from another stream of research also provided convergent evidence that the physiology of appetite and eating behavior differs between men and women⁵⁰⁻⁵⁴. Women have been shown to display stronger brain activation to appetite-enhancing food stimuli, which they also liked more⁵²⁻⁵³. At the same time, women also show a stronger propensity for dietary self-restraint, which is potentially linked to sociocultural and psychological factors, such as perceived pressure to be thin⁵². To reduce these sources of variance, solely women were recruited for the current study."

The experimental design is lacking a control group that would allow for dissecting the mechanisms of positive and negative expectancy on hunger and food valuation. In my view this is not a detrimental flaw as the two conditions investigated (positive and negative expectancy) still allow for the proof-of-concept that expectancy does have an effect. But again, this might be addressed in a more salient way.

We agree with the reviewer that a direct comparison between the decreased- and increased-hunger suggestion groups directly addressed the initial research questions about the effects of expectation on hunger and hunger-addressing economic behavior.

However, to address the reviewer's comment and related concerns raised by reviewers 1 and 3, we collected additional behavioral data from 83 participants who were administered a glass of water without any suggestion about the potential effects of the water on hunger.

Expectancy ratings showed that the control group expected significantly less effects of the water on their hunger than the two suggestion-based groups, which further validated our experimental design. The linear main effects of group on all other dependent variables (hunger ratings, stimulus value, DDM parameters) then showed that the control group (coded as a baseline, 0) was located in between the decreased- (coded -1) and increased- (coded 1) hunger suggestion groups, but often the control group was non-different from the increased hunger suggestion group in posthoc two-sampled comparisons.

We have revised the respective results and figures, accordingly, and have revised the discussion for interpretation of these results (page 18, lines 489–501), which now read:

“Most of the moderation results were specific to participants in the decreased-hunger suggestion group, which also showed differences in the behavioral and computational parameters from the control, no-suggestion group. The increased hunger suggestion group was in many findings non-different from the control, no-suggestion group. It is, therefore, possible that the decreased hunger suggestion group displayed a placebo effect, whereas the increased-hunger suggestion group reflected the natural fluctuation of hunger over the course of the experiment, similar to the control group. However, expectations and hunger ratings were significantly greater in the increased-hunger suggestion group than the control group. On the contrary, the health drift weight influenced the accumulation of evidence toward a food decision similarly in the control and the decreased hunger suggestion groups, and different to the increased hunger suggestion group. More research is therefore needed to characterize the increased-hunger suggestion group relative to the control group at the neural, hormonal, and cognitive levels.”

Rating scales: The wording and dimension of the scales used to assess e.g. expectancy ratings (but also the other behavioural results such as valuation etc) should be understandable in the Results section without having to double-check with the methods. The same applies to all figure legends where the range of potential ratings should be indicated on the y-axis. (e.g. from text to text; 1-10, Likert scale, for Figure 1).

We thank the reviewer for this advice and have added this information to the respective results sections: lines 674–677 for expectancy ratings, lines 680–684 for hunger reports, and lines 691-696 for stimulus value ratings. We have done the same for the figure legends.

Why were different Likert scales (1-7; 1-10) used for the different ratings ?

The hunger ratings were given on a 7-point Likert scale, in accordance with past studies from our lab and that of others (Hare et al. 2009, 2011; Schmidt et al. 2018), which allows comparisons of hunger between studies in the long term. A similar logic was used for the expectancy ratings about the efficiency of the drinks in decreasing or increasing hunger, which were given on a 10-point Likert scale to match other ongoing studies conducted on patients (i.e., Bottemann et al. 2022) aiming to compare expectancy ratings between clinical conditions, treatments, and participants.

Why were no physiological measures such as blood-glucose levels, leptin, ghrelin etc. obtained?

This would have been the perfect study to obtain these valuable measures and relate these to the subjective ratings of hunger and food valuation.

We regret not having been able to do this, but due to logistical and regulatory limits imposed on this study on healthy participants, we could not draw blood to measure these physiological markers of hunger.

However, we would also like to mention that similar suggestion-based placebo effects have already been shown to affect fluctuations in blood ghrelin levels (Hoffmann et al. 2018, Crum et al. 2011). Moreover, although ghrelin levels fluctuate rapidly, within few hours and as a function of food intake and energy stores, leptin blood levels, on the contrary, vary much more slowly and as a function of changes in weight (i.e., the loss or gain of leptin-producing white adipose tissue cells). Therefore, only ghrelin would have been possible and relevant to mark hunger before and after the placebo intervention. Unfortunately, we were not equipped with the necessary tools to perform this analysis.

Other comments:

Neuroimaging: why were different methods used to correct for multiple comparison? E.g. cluster correction was used for results reported on page 4 and following, but peak voxel correction was used for results reported in the supplement.

We apologize for this inconsistency. The moderation of whole brain activation at the time of food choice by expectancy ratings (Figure 1d) survived family-wise error corrections for multiple comparisons at the cluster level, but not at the voxel (peak) level, unlike other contrasts reported in the main text and supplement. Cluster-wise family-wise error (FWE) correction for multiple comparisons across the whole brain is valid, as long as the cluster defining voxel-wise threshold is non-liberal and the samples are well powered (Woo et al. 2014), which was the case in our study (e.g., voxel wise $p < 0.001$ uncorrected for a sample of $n = 28$ in the decreased-hunger suggestion group). We therefore also considered activation that survived cluster-level family-wise error corrections in SPM. To be clearer about the threshold of this random effect moderation of brain activation, we now display voxels that survived the initial uncorrected voxel-wise threshold of $p < 0.001$ with an extended threshold of $k = 44$, which corresponded to $p_{FWE} < 0.05$ cluster-wise control for family wise errors on the whole brain. We added the following lines to the legend of Figure 1:

“(d) Statistical parametric maps of significant group level, random effect moderation of brain activation at the time of food choice by expectations about the drink’s efficiency to decrease hunger. Voxels in yellow are displayed for visualization purposes at an uncorrected threshold of $p < 0.001$, with an extended threshold of $k = 44$ voxels, which corresponded to $p_{FWE} < 0.05$ cluster-wise control for family-wise error across the whole brain. Activation was taken at the local maxima (MNI $x=6$) on the sagittal slice, showing the extent of the activation in mPFC from the superior frontal gyrus to the anterior cingulate cortex. SPMs are superimposed on the average anatomical brain image from 57 participants.”

Given that the definition of ROIs used for small-volume correction is always somewhat arbitrary when selecting coordinates from previous studies I strongly recommend to base SVC on anatomically defined regions. This also makes sure that the size of the region that SVC is based on is realistic (a 10mm sphere is rather small for several of the regions investigated).

We thank the reviewer for this advice. Our dlPFC ROI for small volume correction of the PPI SPMs were defined functionally following a theory-driven approach. To follow the reviewer’s recommendation, we also applied SVC using an anatomically defined dlPFC ROI comprising

Brodmann areas 46 and 9. The activation located in this anatomically defined ROI remain significant. We have added this additional check to the figure legends in the main manuscript and more carefully explain our rationale for using a functionally defined dlPFC ROI pages 36, lines 1005–1023, which read:

“Definition of regions of interest (ROI)

We used a theory-driven approach to determine regions of interest for (1) the mediation analysis and (2) to threshold second-level statistical parametric maps (SPMs) for PPI regressor-related brain activation. Previous work has shown that the decision (action selection) stage of value-based decision-making is implemented by the vmPFC in connection with other prefrontal brain regions, such as the dorsolateral prefrontal cortex (dlPFC)²⁷⁻³¹. Importantly, this work showed that the dlPFC plays a moderating role for value encoding within the vmPFC under cognitive regulation, such as when participants forgo short-term rewarding tasty foods in favor of more abstract and long-term rewarding healthy foods^{19,20,31}. We reasoned that this type of cognitive regulatory success requires more attentional resources, similar to overcoming interference during a Stroop effect. We therefore defined the dlPFC functionally using an independent interference–resolution localizer task, the MSIT task (see SI section 6). In more detail, to extract beta values of this region at time of food choice, and to make small-volume corrections of the PPI SPM, a 10-mm sphere was centered on the dlPFC MNI coordinates ([40, 42, 26]) that were more strongly activated in incongruent than congruent MSIT trials (see SI table 15). Note, the vmPFC–dlPFC PPI connectivity also survived small-volume correction when using an anatomically defined dlPFC ROI that comprised grey matter from Brodmann areas 9 and 46 built using the SPM wfupickatlas.”

Minor:

Please check page 9, lines 255-258. Is there a “with” missing after “scaled”? The sentence is unclear.

Thank you for catching this. We have reformulated the entire paragraph including this sentence, lines 352-358, which now reads:

“We observed a significant positive correlation ($R = 0.42$, $p = 0.02$, $BF_{10} = 3.1$) for participants of the decreased-hunger suggestion group (Figure 3d). Participants in the decreased-hunger suggestion group, who considered the healthiness more during evidence accumulation, were also those who displayed stronger vmPFC – dlPFC PPI connectivity during the decision stage of choice formation. The moderation of vmPFC – dlPFC connectivity by healthiness (relative to tastiness) was non-significant for participants in the increased-hunger suggestion group ($R = -0.008$, $p > 0.05$).”

Was the phase of the menstrual cycle assessed in the participants? And would this allow for exploratory analyses regarding hormonal influences on expectation effects on interoception?

Unfortunately, we did not collect information about the phase of the menstrual cycle. We now acknowledge this limitation in the discussion paragraph about other dimensions of interoception, page 18–19, lines 513–526, which now reads:

“Interoception corresponds to the sensing of bodily states⁴⁶. Here, we used a measurement of interoceptive sensitivity by asking participants to self-evaluate their hunger after self-reported fasting. The question is still open as to the extent hunger can be objectively sensed and whether a person’s beliefs about her hunger can affect other dimensions of interoception, such as behavioral and neural interoceptive accuracy and metacognitive confidence in bodily signal detection. A limitation of our results is also associated with potential biases induced by inter-individual differences in hormonal patterns and self-reported fasting. This study did not collect data on objective hormonal markers of hunger or information about the phase of the participant’s menstrual

cycle. Although similar suggestion-based placebo interventions have been demonstrated to affect ghrelin release^{14,15}, it is still unknown how hormonal variables would have affected hunger-addressing economic behavior and the effects of the suggestion-based placebo interventions. Future studies should explore potential hormonal effects on dietary decision-making under the influence of idiosyncratic higher order beliefs about the body.”

A figure visualizing the experimental protocol with the different time points and tasks would be helpful.

Thank you for this advice. We have added numbers to the various steps of the experimental protocol for clearer reading. Figure 4 is now as shown below:

Figure 4. Experimental procedure. The scheme illustrates the temporal organization of the experiment with the different steps indicated by the numbers 1 to 4. The black panels correspond to computer screenshots of individual events for the hunger and expectancy ratings given before and after fMRI (outside the MRI scanner) and the dietary decision-making task and multi-source interference task performed during fMRI. The duration of the two fMRI tasks is shown in seconds.

Reviewer #3 (Remarks to the Author):

The study investigated how prognostic beliefs about hunger affect appetitive interoceptive experiences and economic behavior by combining a placebo intervention on hunger with

computational modeling and functional magnetic resonance imaging. The results suggest that prognostic beliefs about hunger shape hunger experiences, food valuation, and food-value encoding in the prefrontal cortex. The study also found that taste and health information integration influence the placebo effects on food choices. The findings provide insights into the neurocognitive mechanisms underlying how higher order prognostic beliefs shape non-aversive interoceptive sensitivity and decision-making. The topic is interesting and relevant, the paper is very well written, and the methods appear sound. I enjoyed reading the article. The large sample size is very impressive, especially for a human fMRI study. My enthusiasm is however damped due to critical methodological issues.

Thank you for your positive feedback.

My main concern is the missing control group. Half of the participants were told that the water drink would increase hunger, whereas the other half was told the opposite. Hunger ratings increased in both groups – which was to be expected considering the time participants stayed without food prior to the experiments. Prognostic beliefs about hunger for each group were different from one (one-sample t-test), suggesting that each intervention had an effect different from “no expected efficiency”. But, considering participants’ generally increasing hungriness, couldn’t comparable prognostic beliefs about hunger not also stem from individuals who received water without suggestions?

To address your concern, we collected behavioral data from a new sample of 83 participants. These participants were matched for age, body composition, and level of education to the two hunger-suggestion groups. Participants of the control group were administered a glass of water prior to performing the dietary decision-making task without receiving any suggestion or information about the water’s ingredients or potential effect on hunger.

We found that general expectancy ratings (i.e., how much they expected the drink to affect their hunger) varied between the 83 control participants, but on average were significantly lower than those of the decreased- and increased-hunger suggestion groups. These findings indicate that the placebo intervention induced stronger prognostic beliefs about hunger than those found in the control group. Moreover, we also found that the increase in hunger was greater in the increased-hunger suggestion group than the control group and lower in the decreased-hunger suggestion group than in the control group. We therefore infer that the increases in hunger do not stem solely from the control participants.

There were also no between-group effects, questioning whether the two interventions really induced different beliefs about hunger.

We apologize for being unclear on this non-difference in the strength of expectancy in the two hunger-suggestion groups, which was an important requirement for the interpretation of all subsequent group differences.

It is important to note that expectations were rated on the same scale from 1, minimum expected efficiency, to 10. We have added a verbatim formulation of the expectancy measurement to the methods section in the main manuscript (page 24, lines 674-677) and present the expectancy ratings

in terms of the anticipated efficiency of the drink on the participants hunger in the figure legends and corresponding results section.

In more detail, expectations were framed differently for the two hunger-suggestion groups. The framing corresponded to subjective anticipation of the drink's efficiency to decrease hunger in the decreased-hunger suggestion group or increase hunger in the increased-hunger suggestion group. We therefore argue that this difference in the framing of the participants expectations was sufficient to induce differentially valenced beliefs about the anticipated effects of the drink on hunger that were, however, comparable in strength.

The additional behavioral data obtained for the control participants, who drank the water without any suggestion about its potential effect on hunger and expected significantly less strong effects on hunger relative to the two hunger-suggestion groups, further support our findings.

Although significant correlation analysis and model parameter comparison revealed distinct differences between both intervention groups, it remains questionable whether any effect in each of the two groups would be different from a group where water, but no suggestions have been given? Without such a control group, it is impossible to say in which way suggestions affected participants' beliefs.

We re-conducted our data analyses including the observed behavioral and computational variables of the control, no-suggestion group participants. The results of these analyses are consistent with our initial findings and are further detailed in response to your next comment below concerning the computational aspects and in the corresponding results sections.

Briefly, the control group differed in terms of the magnitude (strength) of the expected effect from the water on hunger relative to the two hunger-suggestion groups and their hunger also increased more than that of the decreased-hunger suggestion group and less than that of the increased-hunger suggestion group.

We observed the same pattern of results for food valuation and the computational parameters, with significant linear main effects of group. However, post-hoc tests showed that this linear effect of group was driven by significant differences between the decreased-hunger suggestion group and the no-suggestion control group, except for the initial starting bias, which was non-different between control and decreased hunger suggestion group. For many other variables of interest (e.g., stimulus value, health and taste drift weights) the increased-hunger suggestion group did not significantly differ from the control group. A non-difference is difficult to interpret, and we added the following lines concerning this finding in the discussion section page 18, lines 489–501, which reads:

“Most of the moderation results were specific to participants in the decreased-hunger suggestion group, which also showed differences in the behavioral and computational parameters from the control, no-suggestion group. The increased hunger suggestion group was in many findings non-different from the control, no-suggestion group. It is, therefore, possible that the decreased hunger suggestion group displayed a placebo effect, whereas the increased-hunger suggestion group reflected the natural fluctuation of hunger over the course of the experiment, similar to the control group. However, expectations and hunger ratings were significantly greater in the increased-hunger suggestion group than the control group. On the contrary, the health drift weight influenced the accumulation of evidence toward a food decision similarly in the control and the decreased hunger suggestion groups, and different to the increased hunger suggestion group. More research is therefore needed to characterize the increased-hunger suggestion group relative to the control group at the neural, hormonal, and cognitive levels.”

Why did authors apply only tDDM and sDDM and not also other model variants? The modeling approach was adapted from Maier et al. (Ref. 30). Maier et al. applied tDDM and sDDM, but also four other variants (see Suppl. Tab. 1 in Ref. 30). They then used BIC, instead of DIC, to identify the best fitting model. In the paper under consideration, authors argue that tDDM has been validated by two independent studies (Ref. 30 & 52). However, model fit may vary across samples and designs. The finding that tDDM, as the more “context-sensitive” model, outperformed sDDM is also not surprising.

We thank the reviewer for steering us to verify and further challenge our modeling approach.

The rationale for using DICs instead of BICs and AICs for model selection was the following: The drift rates were specified by a stochastic Wiener process implemented in JAGS, which is different from the approach reported by Maier et al. (2020). The model was then estimated using hierarchical Gibbs sampling and an MCMC method, which was also a different estimation strategy compared to Maier et al. 2020. For MCMC outputs of hierarchical Bayesian models the deviance information criterion (DIC) was proposed by Spiegelhalter et al. (2002) and Gelman et al. (2013) as a standard criterion for Bayesian model selection and corresponds to a generalization of the AIC.

Furthermore, the hierarchical estimation approach is based on an average drift rate and, thus, cannot explicitly distinguish between when and how often health and taste information affected the drift rate. The relative timing parameter thus reflected a more general form of temporal dynamics during the decision stage. A stepwise approximation of the drift rate following the approach used by Maier et al. 2020 would allow looking into the fits of other variants of the tDDM.

Although it was not the initial scope of the study to distinguish different time-varying DDMs, we have looked into this question more deeply. We re-fitted these alternative time-varying DDMs using the stepwise approximation of the drift rate implemented by the Rcpp package in R, and following the parameter estimation used by Maier et al. 2020. However, parameter recovery for the various time-varying versions of the DDM (start, stop, and start&stop DDMs) was very poor for the relative time parameter in every version of the time-varying DDM. The poor parameter recovery indicated that the time-varying DDM was not the most “context-sensitive” model for this dataset, and did not give parameters that described the data better than any other set of parameters. This sanity check led us to reconsider our modeling approach entirely. We decided to model our data with a standard, non-time-varying DDM. In more detail, we compared the fits between a standard DDM, which assumed that participants used taste and health information to decide, against two alternative standard DDMs, which assumed that participants solely used taste or health, respectively. The winning model was the standard 2 separate taste and health drift weight DDM, which provided identifiable parameters. The parameter recovery was now excellent for all free parameters of the model (see SI Table 13, SI Figure 7), and we are confident that this model is most relevant for drawing inferences on how suggestions about hunger influenced the dynamics of the decision stage.

We have modified the corresponding results section and figure 3, entirely. The results show that the suggestions about hunger influenced the initial starting bias toward accepting food (a ‘yes’ food choice), and how much participants weighed taste and health information during choice formation.

The findings do not change the basic inferences we can draw from the computational analyses. Notably, these computational modeling results provide direct evidence that participants are biased and weigh information about food as a function of their hunger expectation and hunger state. We argue that these effects reveal cognitive mechanisms at work that mediate the suggestion-based placebo effects, further corroborated by our ROI-based mediation results and the moderation of

dIPFC–vmPFC PPI by the relative taste and health drift weights in the decreased hunger suggestion group.

More work is needed to characterize the temporal dynamics. As discussed in the initial submission, we reason that other measurement approaches, such as for example EEG, have a better temporal resolution to determine the temporal dynamics of suggestion-based placebo effects on attentional filtering mechanisms.

Was the data tested for normal distribution?

Hunger ratings at baseline and those at end of the experiment and the difference in hunger ratings between baseline and the end of the experiment were not normally distributed for either hunger-suggestion group, as indicated by significant Shapiro-Wilk tests (decreased-hunger suggestion group, $p = 0.037$, $p = 0.013$, and $p < 0.001$, respectively, $N = 88$; increased-hunger suggestion group: $p = 0.005$, $p = 0.032$, and $p < 0.001$, respectively, $N = 84$). However, the main effects tests and t-tests should be valid given the large sample size (Fagerland M BMC 2012).

Moreover, we now also report Bayes factors (BF_{10}) from the Bayesian t-tests for the main comparisons between suggestion groups for all observed behavioral and computational variables.

When first reading the Introduction, I was expecting a main effect test applied to the entire sample followed by post-hoc tests for each sub-group (increase and decrease hunger suggestion group).

Thank you for catching this. Although, we initially conducted main effects tests for the entire sample for the hunger ratings, we also tested for main effects on expectancy and stimulus value ratings. We have added these results to the relevant results sections, page 5 (lines 131–136 for expectations), page 7 (lines 181–194 for hunger ratings), and page 8 (lines 210–214 for stimulus value ratings). We took this opportunity to unify our statistical analysis for these three behavioral measures of interest by fitting linear mixed effects models, which allowed for the testing of nested between- (i.e., suggestion) and within-group (measurement time) main effects and controlling for the between-participant variable BMI (see our response to your comment on potential confounders induced by differences in BMI below).

In the Introduction, authors should better motivate their hypotheses.

We thank the reviewer for this relevant comment on the introduction. We took this as an opportunity to reformulate it more towards the main questions and hypothesis from lines 61 to 69. We also took the advice to clarify specific aspects of the intervention combining the placebo and verbal suggestion to more clearly delineate the scope of this work more towards basic research on the neurocognitive mechanisms of suggestion induced placebo effects on appetitive domains of behavior and judgement.

Although the total sample size is impressive, the effect sizes for many tests appear rather weak. Authors should provide sample and effect sizes for each test.

Thank you for this advice. To support our results, we have added Cohen's d values to the main results, which indicated moderate to strong effect sizes that ranged between a minimum of $d = 0.3$ and a maximum of $d = 0.6$. Moreover, we would like to mention that our results replicated previous work on placebo effects on subjective hunger ratings, and this robust effect of hunger suggestion on hunger ratings provided a solid basis for this study's scope in identifying the underlying neural and

computational mechanisms. Finally, observed differential brain activation between the suggestion groups was family-wise error corrected at the voxel and cluster level for the entire brain, which further provides evidence for non-weak effects.

What was the rationale behind applying the cluster instead of the voxel level threshold to most contrast images (and the voxel-level to only some)? Applying the cluster level to functional SPMs is unconventional since it can lead to a loss of spatial resolution, introduce bias in the analysis, and increase the likelihood of false positives or false negatives.

Whole brain family-wise error (FWE) correction at the cluster level is a valid correction method for family wise errors of task-based fMRI activations. We used it to localize activation in brain regions that have been well reported in the literature to underly value-based decision-making, such as the vmPFC, and its moderation by cognitive regulation, such as by the dlPFC. In this perspective and since we were not seeking more specific locations within these well-known brain regions of economic behavior, the use of cluster-wise corrected whole brain thresholds appeared to be reasonable. Moreover, the cluster defining primary threshold was non-liberal (i.e., $p < 0.001$), and we used a well-powered sample size for fMRI following recommendations provided by Woo et al. 2014. Please note that the main findings also survived family-wise error correction for multiple comparisons at the voxel level. We have now added this information to the relevant methods section and figure legends.

Why only female participants? I understand authors' argument of gender-related dietary self-restraint, but sex hormonal responses in women are known to interfere with the valuation of food pictures, which may vary over menstrual cycle. What about contraceptives and their potential influence? Many previous reward studies tested only male participants because of these known sex hormonal interactions in women.

Previous studies have shown a stronger physiological response in women to placebo interventions in the pain (Colloca et al., 2016; Theysohn et al., 2014) and hunger domain (see Hoffmann et al. 2018). Moreover, a large body of research has provided evidence that the physiology of appetite differs between men and women (Davy et al., 2006; Marino et al., 2011). For example, women were reported to show stronger brain activation to appetite-enhancing food stimuli than men (Frank et al., 2010; Killgore & Yurgelun-Todd, 2010). Finally, we also restricted inclusions to female participants because of reported differences in dietary restraint/impulsivity (Rolls et al. 1991). We have now clarified this in the main text page 21, lines 578–591, which read:

“Inclusion was restricted to self-identified female participants for the following reasons. Studies of placebo effects on pain have provided robust evidence that women display higher levels of positive expectations and greater placebo effects than men⁴⁷⁻⁴⁹. Importantly, we used a suggestion-based placebo intervention on hunger from a previous behavioral study of placebo effects on objective (i.e., hormonal) and subjective (i.e., self-reports) markers of hunger¹⁵. A secondary finding of this study was a significantly stronger placebo effect for women than men, which further motivated the recruitment of women for the current study. Empirical findings from another stream of research also provided convergent evidence that the physiology of appetite and eating behavior differs between men and women⁵⁰⁻⁵⁴. Women have been shown to display stronger brain activation to appetite-enhancing food stimuli, which they also liked more⁵²⁻⁵³. At the same time, women also show a stronger propensity for dietary self-restraint, which is potentially linked to sociocultural and psychological factors, such as perceived pressure to be thin⁵². To reduce these sources of variance, solely women were recruited for the current study.”

To address the reviewer's comment more specifically, we did not collect information about the menstrual cycle and contraceptives and, therefore, cannot test whether these covariates could have

potentially influenced food valuation. We now mention this limitation for the interpretation of our findings in the discussion section page 19, lines 518–526, which reads:

“A limitation of our results is also associated with potential biases induced by inter-individual differences in hormonal patterns and self-reported fasting. This study did not collect data on objective hormonal markers of hunger or information about the phase of the participant’s menstrual cycle. Although similar suggestion-based placebo interventions have been demonstrated to affect ghrelin release^{14,15}, it is still unknown how hormonal variables would have affected hunger-addressing economic behavior and the effects of the suggestion-based placebo interventions. Future studies should explore potential hormonal effects on dietary decision-making under the influence of idiosyncratic higher order beliefs about the body. ”

I couldn’t find any information about participants’ body weight. Did authors assess the BMI? Have all participants been lean? Anorexia, overweightness, and any form of obesity would represent highly influential factors on task performance and fMRI signals. Was the BMI (and age) counterbalanced across groups?

Information about body composition (weight, BMI, %fat) is provided in SI table 9. The average BMI was approximately 23kg/m² but ranged from 15 to 51. The suggestion groups did not differ in terms of BMI ($t(170) = 0.33, p = 0.73$), body fat ($t(166) = 0.66, p = 0.51$), age ($t(170) = 1.49, p = 0.14$), or level of higher education ($t(169) = -0.54, p = 0.59$). Nor did they differ from the control group (decreased vs control: $t(169) = 1.65, p = 0.10$ for BMI, $t(165) = 0.31, p = 0.76$ for body fat, $t(169) = 1.70, p = 0.09$ for age, $t(169) = 0.97, p = 0.33$ for years of higher education; increased vs control: $t(165) = 1.51, p = 0.13, t(165) = -0.06, p = 0.95$ for body fat, $t(165) = 0.17, p = 0.87$ for age, $t(164) = 1.56, p = 0.12$ for years of higher education).

We reconducted all of our main effects tests on expectancy, hunger, and stimulus value ratings controlling for BMI by adding BMI as a fixed effects regressor to the LMEs.

To ensure that overweight and underweight participants did not drive effects, we re-conducted the analyses solely for participants with a normal BMI ($18.5 < \text{BMI} < 30$). The main effect of the measurement time point ($\beta_{\text{testing time}} = 0.38 \pm 0.04, t(427) = 10.23, p < 0.001, 95\% \text{ CI } [0.31 - 0.46]$) and the suggestion group by measurement time point interaction ($\beta_{\text{group*testing time}} = 0.21 \pm 0.05, t(427) = 4.49, p < 0.001, 95\% \text{ CI } [0.12 - 0.30]$) on hunger remained significant. This was also true for the main effect of suggestion group on the stimulus value ($\beta_{\text{group}} = 0.10 \pm 0.04, t(213) = 2.90, p = 0.004, 95\% \text{ CI } [0.03 - 0.18]$).

Next, we divided participants into three weight groups: underweight ($\text{BMI} < 18.5$), normal weight ($18.5 < \text{BMI} < 25$), and overweight and obese ($\text{BMI} > 25$), and fitted the following linear mixed effects model (LME) to the hunger ratings and stimulus value ratings:

1. $\text{hunger} \sim \text{Intercept} + \text{suggestion group} + \text{measurement time} + \text{BMI} + \text{weight group} + \text{suggestion group} * \text{measurement time} + (\text{Intercept} \mid \text{participant number})$
2. $\text{stimulus value} \sim \text{Intercept} + \text{suggestion group} + \text{BMI} + \text{weight group} + \text{suggestion group} * \text{weight group} + (\text{Intercept} \mid \text{participant number})$

For the hunger ratings we found a main effect of suggestion group ($\beta_{\text{group}} = 0.21 \pm 0.08, t(503) = 2.59, p = 0.01, 95\% \text{ CI } [0.05 - 0.38]$), a main effect of measurement time point ($\beta_{\text{testing time}} = 0.35 \pm 0.04, t(503) = 9.96, p < 0.001, 95\% \text{ CI } [0.28 - 0.42]$), and a significant suggestion group by measurement

time interaction ($\beta_{\text{suggestion group*time}} = 0.46 \pm 0.22$, $t(503) = 2.06$, $p = 0.04$, 95% CI [0.02 – 0.89]), and no effect of weight group ($\beta_{\text{weight group}} = 0.12 \pm 0.19$, $t(504) = 0.62$, $p = 0.5351$, 95% CI [-0.27 – 0.51]).

We also redetected the main effect of suggestion group on stimulus value ($\beta_{\text{group}} = 0.11 \pm 0.03$, $t(250) = 3.04$, $p = 0.003$, 95% CI [0.03 – 0.18]) but no effect of weight group ($\beta_{\text{weight group}} = -0.03 \pm 0.08$, $t(250) = -0.41$, $p = 0.68$, 95% CI [-0.19 – 0.12]) or suggestion group by weight group interaction ($\beta_{\text{group*weight group}} = -0.05 \pm 0.06$, $t(250) = -0.75$, $p = 0.46$, 95% CI [-0.16 – 0.07]).

This further shows that our results cannot be explained by differences in BMI.

Reviewer #2 (Remarks to the Author):

The authors provide a revised manuscript that thoroughly addresses the points raised by the reviewers. The authors convincingly explain the choices made regarding methods and analyses. The study significantly advances the field, and I look forward to its publication.

Reviewer #3 (Remarks to the Author):

The authors did an excellent job addressing my criticisms. Especially the comparison of the two suggestion groups with the new control group and the revision of the DDM approach convinced me very much.

I have only two small remaining points:

1. fMRI methods: I think the authors should briefly explain in one sentence why they specifically corrected for dropouts in the OFC region although the OFC was not part of the hypotheses.
2. the abbreviation DIC should be written out in line 312.

Dear reviewers

Thank you for reviewing our revised manuscript “Taste matters: Mapping expectancy-based appetitive placebo effects onto the brain” (NCOMMS-23-06025A).

We hereby submit a final revision of the manuscript to address your advice. Please find below our responses.

Sincerely,
Dr Liane Schmidt on behalf of all co-authors

REVIEWERS' COMMENTS

Reviewer #2 (Remarks to the Author):

The authors provide a revised manuscript that thoroughly addresses the points raised by the reviewers. The authors convincingly explain the choices made regarding methods and analyses. The study significantly advances the field, and I look forward to its publication.

Thank you.

Reviewer #3 (Remarks to the Author):

The authors did an excellent job addressing my criticisms. Especially the comparison of the two suggestion groups with the new control group and the revision of the DDM approach convinced me very much.

Thank you.

I have only two small remaining points:

1. fMRI methods: I think the authors should briefly explain in one sentence why they specifically corrected for dropouts in the OFC region although the OFC was not part of the hypotheses.

We have added the following sentence to the methods section, page 26

“This correction for signal dropout was applied because the ventromedial prefrontal cortex, a brain region of interest, encompasses the medial portions of the orbitofrontal cortex.”

2. the abbreviation DIC should be written out in line 312.

We have written out DIC in line 312 – Deviance Information Criterion